# Schur sector of Argyres-Douglas theory and $W$-algebra

**Dan Xie[1,2] and Wenbin Yan[1]⋆**

**1** Yau Mathematics Science center, Tsinghua University, Beijing, 10084, China
**2** Department of Mathematics, Tsinghua University, Beijing, 10084, China

⋆ wbyan@mail.tsinghua.edu.cn

## Abstract

We study the Schur index, the Zhu's $C_2$ algebra, and the Macdonald index of a four dimensional $\mathcal{N}=2$ Argyres-Douglas (AD) theories from the structure of the associated two dimensional $W$-algebra. The Schur index is derived from the vacuum character of the corresponding $W$-algebra and can be rewritten in a very simple form, which can be easily used to verify properties like level-rank dualities, collapsing levels, and S-duality conjectures. The Zhu's $C_2$ algebra can be regarded as a ring associated with the Schur sector, and a surprising connection between certain Zhu's $C_2$ algebra and the Jacobi algebra of a hypersurface singularity is discovered. Finally, the Macdonald index is computed from the Kazhdan filtration of the $W$-algebra.

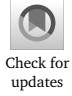
---

# 1   Introduction

It is important to understand moduli spaces of vacua of four dimensional (4d) $\mathcal{N} = 2$ superconformal field theories (SCFTs). An $\mathcal{N} = 2$ SCFT could have a Coulomb branch and a Higgs branch. The low energy effective theory on the Coulomb branch is solved by finding a Seiberg-Witten geometry. Almost every nontrivial 4d $\mathcal{N} = 2$ SCFT has a Coulomb branch, which is parameterized by expectation values of half-BPS operators $\mathcal{E}_{r,(0,0)}$[1]. These operators form a ring which is freely generated for almost all the theories we know[2]. The important question is to determine the rational number $r$ of each Coulomb branch operator $\mathcal{E}_{r,(0,0)}$. In practice, one can often easily determine them using the Seiberg-Witten (SW) geometry.

It is also possible for a 4d $\mathcal{N} = 2$ SCFT to have a Higgs branch, which is parameterized by expectation values of half-BPS operators $\hat{\mathcal{B}}_R$. These operators form a nontrivial ring called the Higgs branch chiral ring. Unlike the common appearance of the Coulomb branch, not all $\mathcal{N} = 2$ SCFT has a Higgs branch and in fact there does exist a large class of $\mathcal{N} = 2$ SCFTs which do not have a Higgs branch.

Given the asymmetry between the Higgs branch and the Coulomb branch, one might wonder whether a protected sector could exist for all non-trivial $\mathcal{N} = 2$ SCFT and contains the Higgs branch when the theory has one. Such sector indeed exists and is called the Schur sector [3, 4], which contains Higgs branch operators $\hat{\mathcal{B}}_R$ and operators $\hat{\mathcal{C}}_{R,(j_1,j_2)}$. It is in general quite difficult to determine this sector as there is no powerful tool as the SW geometry of the Coulomb branch.

The understanding of the Schur sector becomes possible because of the following 4d/2d correspondence found in [5] (see [6–56] for further developments). There is a map between the Schur sector of a 4d $\mathcal{N} = 2$ SCFT and a 2d vertex operator algebra (VOA). Once the 2d VOA for a 4d $\mathcal{N} = 2$ SCFT is identified, one can learn a lot about the Schur sector of the 4d theory from known properties of 2d VOA.

---

[1] $\mathcal{N} = 2$ SCFT has a bosonic symmetry group $SO(2,4) \times SU(2)_R \times U(1)_R$, and the highest weight representation is labeled as $|\Delta, R, r, j_1, j_2\rangle$, here $\Delta$ is the scaling dimension, $R$ labels the $SU(2)_R$ representation, $r$ is $U(1)_R$ charge, and $j_1, j_2$ are left and right spins. Short supermultiplets are classified in [1], and there are three types of half BPS operators which are important to us: a): $\mathcal{E}_{r,(0,0)}$ with $\Delta = r$ and $R = 0$; b): $\hat{\mathcal{B}}_R$ with $\Delta = 2R$, $r = j_1 = j_2 = 0$; c): $\hat{\mathcal{C}}_{R,(j_1,j_2)}$ with $\Delta = 2 + 2R + j_1 + j_2$ and $r = j_2 - j_1$.

[2] See [2] for the discussion on the possibility of nontrivial Coulomb branch chiral ring.

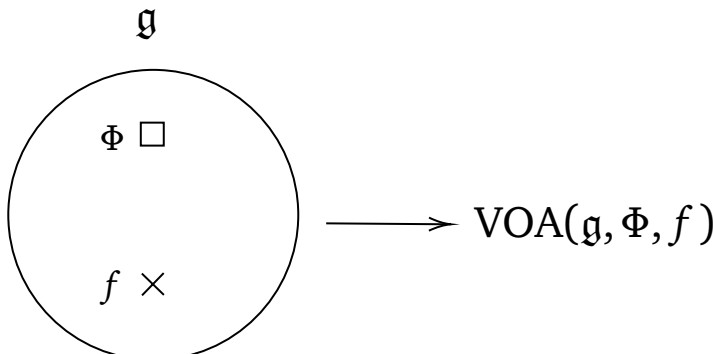

Figure 1: A mapping of a 6d $(2,0)$ configuration to a 2d VOA, here $\mathfrak{g}$ is a simple Lie algebra, $\Phi$ is an irregular singularity, and $f$ represents a regular singularity. If $\Phi$ is of principal nilpotent type with type $h^\vee$ and an integer label $k$ [39, 58, 59], the VOA is just $W$-algebra $W^{k'}(\mathfrak{g}, f)$ with $k' = -h^\vee + \frac{h^\vee}{h^\vee + k}$.

## Summary of results

For a large class of 4d $\mathcal{N}=2$ Argyres-Douglas type SCFTs engineered from 6d $(2,0)$ theories, we have identified their associated 2d VOAs as $W$-algebras $W^{k'}(\mathfrak{g}, f)$ [17, 29, 57] shown in figure 1. Such $W$-algebra is derived from the quantum Drinfeld-Sokolov (qDS) reduction of an affine Kac-Moody (AKM) algebra $V^{k'}(\mathfrak{g})$ with level $k'$, and $f$ is a nilpotent element of Lie algebra $\mathfrak{g}$. In [57], the corresponding 4d theory for $W^{k'}(\mathfrak{g}, f)$ has been constructed for any simple Lie algebra $\mathfrak{g}$ with arbitrary nilpotent element $f$ of $\mathfrak{g}$ at a level $k' = -h^\vee + \frac{h^\vee}{h^\vee + k}$ which is also been called boundary admissible level [3] , and we will call the corresponding 4d $\mathcal{N}=2$ SCFT $\mathcal{T}[W^{k'}(\mathfrak{g}, f)]$ in this paper.

Such $W$-algebra has been studied in physics and mathematics literature extensively [60–62]. The purpose of this paper is to extract important information of the Schur sector of the 4d theory $\mathcal{T}[W^{k'}(\mathfrak{g}, f)]$ for **any simple Lie algebra** $\mathfrak{g}$ and **any** nilpotent orbit $f$ at a **boundary admissible level** $k' = -h^\vee + \frac{h^\vee}{h^\vee + k}$ from the knowledge of the 2d VOA. We obtain three main results:

1. The Schur index can be computed from the vacuum character of the $W$-algebra $W^{k'}(\mathfrak{g}, f)$, and the character at the boundary admissible level [63] can be written as a product of theta functions. We discovered that the index can be put as a very simple form in terms of plythestic exponential (PE)

$$\mathcal{I}_{W^{k'}(\mathfrak{g}, f)}(q, z) = PE\left[\frac{\sum_j q^{1+j}\chi_{R_j}(z) - q^{h^\vee + k}\sum_j q^{-j}\chi_{R_j}(z)}{(1-q)(1-q^{h^\vee + k})}\right]. \tag{1}$$

Here for a nilpotent element $f$, one has an associated $\mathfrak{sl}_2$ triple and the associated Lie group $G$ of $\mathfrak{g}$ has a subgroup $SU(2) \times G_F$ with $G_F$ being the flavor symmetry group corresponding to $f$. The adjoint representation of $\mathfrak{g}$ decomposes as $adj_{\mathfrak{g}} \to \oplus_j V_j \otimes R_j$ under the subgroup $SU(2) \times G_F$, here $j$ is the spin $j$ representation of $SU(2)$ subgroup, and $R_j$ is the representation under the flavor group $G_F$. $\chi_{R_j}(z)$ is the character of representation $R_j$ of flavor group $G_F$. This formula is a generalization of $\mathfrak{g} = ADE$ case considered in [29] to the arbitrary simple Lie algebra. Though equivalent to the product formula in [63], this simple PE form makes many highly nontrivial properties of 4d and 2d theories **manifest**, i.e. level-rank dualities and collapsing levels of 2d VOA, and more interestingly

---

[3]Here $\mathfrak{g}$ is a simple Lie algebra, $h^\vee$ is its dual Coxeter number and k is an integer with following constraints: a) $h^\vee + k \geq 2$; b) $k$ and $h^\vee$ coprime; c) and $k \neq 2n$ for $\mathfrak{g} = B_N, C_N, F_4$, and $k \neq 3n$ for $\mathfrak{g} = G_2$.

the S-duality conjecture proposed in [64, 65]. This form is also extremely helpful for extracting informations on generators and null states of the VOA.

2. For a VOA, one can define a commutative and associative algebra called the Zhu's $C_2$ algebra [66]. The reduced ring from the $C_2$ algebra is identified with the Higgs branch chiral ring [29, 31, 67]. For $W^{-h^\vee + \frac{h^\vee}{h^\vee + k}}(\mathfrak{g}, f)$ VOA considered in this paper, the Zhu's $C_2$ algebra has a simple form implied by the simple PE form of the vacuum character and we gave a general proposal for its structure. Especially if the 4d theory has **no flavor symmetry**, we conjecture that the $C_2$ algebra is actually **isomorphic** to a Jacobi algebra associated with a quasi-homogenous hypersurface singularity. Although Zhu's $C_2$ can be thought as the associated ring of the Schur sector, its relation with 4d theories is still unclear and should be explored in the future. The Zhu's $C_2$ algebra may be more important than its reduced version for 4d physics because not all 4d $\mathcal{N} = 2$ SCFTs have Higgs branches, while even theories with no Higgs branch (the reduced $C_2$ algebra) have a nontrivial Schur sector which corresponds to Zhu's $C_2$ algebra.

3. To compute the Macdonald index of a given 4d theory, one need to introduce another grading or filtration in the corresponding VOA. For our $W$-algebra, there is a natural filtration called the Kazhdan filtration [68] which we use to define the Macdonald index of the 4d theory. This filtration agrees with the filtration of theories considered in [22], and gives a natural generalization to general models considered in this paper.

This paper is organized as follows: section 2 reviews the basic correspondence between the Schur sector of 4d theory and 2d VOA. Section 3 reviews known results between 4d Argyres-Douglas theories engineered from 6d $(2, 0)$ theories and their associated 2d $W$-algebras. Section 4 studies the Schur index from the vacuum character of the $W$-algebra. Section 5 studies the Zhu's $C_2$ algebra which might be thought of as a ring associated with the Schur sector. Section 6 introduces the Kazhdan filtration of our $W$-algebra and it is used to define the Macdonald index. Finally a conclusion is given in section 7.

## 2 Schur sector and VOA

The representation theory of a 4d $\mathcal{N} = 2$ SCFT was studied in [1]. $\mathcal{N} = 2$ SCFT has a bosonic symmetry group $SO(2, 4) \times SU(2)_R \times U(1)_R$, and the highest weight representation is labeled as $|\Delta, R, r, j_1, j_2\rangle$, here $\Delta$ is the scaling dimension, $R$ labels the $SU(2)_R$ representation, $r$ is $U(1)_R$ charge, and $j_1, j_2$ are left and right spins. Short supermultiplets are classified in [1], and there are three types of half BPS operators which are important to us: a): $\mathcal{E}_{r,(0,0)}$ with $\Delta = r$ and $R = 0$; b): $\hat{\mathcal{B}}_R$ with $\Delta = 2R$, $r = j_1 = j_2 = 0$; c): $\hat{\mathcal{C}}_{R,(j_1,j_2)}$ with $\Delta = 2 + 2R + j_1 + j_2$ and $r = j_2 - j_1$. We are interested in so-called Schur sector which contains operators satisfying the following condition

$$
\begin{aligned}
\frac{1}{2}(\Delta - (j_1 + j_2)) - R &= 0, \\
r + j_1 - j_2 &= 0.
\end{aligned}
\tag{2}
$$

The Schur operators are contained in supermultiplets $\hat{\mathcal{C}}_{R,(j_1,j_2)}$, $\hat{\mathcal{B}}_R$, $\mathcal{D}_{0(0,j_2)}$ and $\bar{\mathcal{D}}_{0(j_1,0)}$. $\mathcal{D}_{0(0,j_2)}$ and $\bar{\mathcal{D}}_{0(j_1,0)}$ multiplets will not appear in theories considered here [69]. $\hat{\mathcal{C}}_{0,(0,0)}$ is the supercurrent multiplet and $\hat{\mathcal{B}}_R$ multiplets contain Higgs branch operators. Notice that for $\hat{\mathcal{C}}$ type supermultiplet, the Schur operator is not the bottom component.

The Macdonald index and the Schur index [3,4] are non-zero in the Schur sector only. The Macdonald index of a $\hat{\mathcal{C}}_{R,(j_1,j_2)}$ multiplet is[4]

$$\mathcal{I}^M_{\hat{\mathcal{C}}_{R,(j_1,j_2)}}(q,T) = \frac{q^{2+R+j_1+j_2}T^{1+R+j_2-j_1}}{1-q}, \tag{3}$$

where $\frac{1}{1-q}$ represents the contribution from derivatives. The Schur index of the same multiplet is given by setting $T = 1$ in above formula

$$\mathcal{I}^{Schur}_{\hat{\mathcal{C}}_{R,(j_1,j_2)}}(q) = \frac{q^{2+R+j_1+j_2}}{1-q}. \tag{4}$$

Moreover, if the theory has a flavor symmetry, one may also add flavor fugacities in both indices, which keep track of the action of the flavor group. Such fugacities are crucial when considering modular properties of indices. Another important property is that Higgs branch operators $\hat{\mathcal{B}}_R$ form a ring and in most cases there is also a Hyperkhaler metric associated with this ring.

## 2.1 Quasi-lisse VOA

VOA arises as the chiral part of a two dimensional conformal field theory. Here we review the mathematical definition of a VOA. A vertex algebra is a vector space $V$ with following properties ($V$ can be thought of as the vacuum module of the chiral part of 2d CFT) [70]:

- A vacuum vector $|0\rangle$.

- A linear map
$$Y : V \to \mathcal{F}(V), \quad a \to Y(a,z) = \sum_n a_n z^{-n-1} = a(z), \tag{5}$$
  where $a_n \in End(V)$. This is just the state-operator correspondence[5]. Given a field $a(z)$, one can recover the corresponding state $|a(z)\rangle = \lim_{z\to 0} a(z)|0\rangle$.

For our purpose, we need to consider the VOA with a conformal vector $\omega$, which is nothing but the chiral part $T(z)$ of the stress tensor. The modes in the expansion of $T(z) = \sum L_n z^{-n-2}$ satisfy the Virasoro algebra (using the standard contour integral and OPE of $T(z)$)

$$[L_n, L_m] = (n-m)L_{n+m} + \frac{c(n^3-n)}{12}\delta_{n+m,0}. \tag{6}$$

The normal order product of two fields $a(z)$ and $b(z)$ is denoted as $: ab : (z)$, and its modes are

$$(: ab : (z))_n = \sum_{n \le -h_a} a_n b_{m-n} + \sum_{n > -h_a} b_{m-n} a_n. \tag{7}$$

In current convention we have $h_a = 1$. Other properties of VOA can be found in [70].

Now let us review the definition of some special VOAs. A VOA is called **rational** if

1. V has finite number of irreducible representations $M_j$.

2. The normalized character $\text{ch}_j = \text{tr}_{M_j}(e^{2\pi i\tau(L_0-\frac{c}{24})})$ converges to a holomorphic function on upper half plane $\mathbb{C}^+$[6].

---

[4]We use the notation $\hat{\mathcal{B}}_{R+1} = \hat{\mathcal{C}}_{R,(-\frac{1}{2},-\frac{1}{2})}$.

[5]In physics literature, the mode expansion of a field takes the form $\sum a_n z^{-n-h}$ with $h$ the scaling dimension. In VOA literature, however, they use above convention of mode expansion so that they can consider VOA without the definition of scaling dimension.

[6]We use ch to denote the character of a VOA and $\chi$ to denote the character of a finite Lie algebra.

3. The function $\mathrm{ch}_j$ span a $SL_2(Z)$ invariant space.

A VOA is called **finitely strongly generated** if there is finite number of elements $a_i \in V$, $i = 1, \ldots, s$ such that the whole VOA is spanned by following normal order products

$$: \partial^{k_1} a_1 \ldots \partial^{k_s} a_s : . \tag{8}$$

Notice that the choice of generators may not be unique and in general there are relations between the above basis. It is interesting to find a minimal generating set of a finitely strongly generated VOA.

For a VOA $V$, there exists a Li's filtration [71] which is a decreasing filtration

$$F^0 \supset F^1 \supset F^2 \supset \ldots, \tag{9}$$

in which each $F^p$ is spanned by following states

$$F^p(V) = \{a^{i_1}_{-n_1-1} a^{i_2}_{-n_2-1} \ldots |0\rangle, \quad \sum n_i \geq p\}, \tag{10}$$

then there is a graded sum of VOA

$$Gr(V) = \oplus_p \frac{F^p}{F^{p+1}}. \tag{11}$$

It is obvious that $F^0 = V$, and $F^1$ is generated by $\{a_{-2}b|a \in V, \ b \in V\}$. Zhu's $C_2$ algebra is defined as [72]

$$R_V = \frac{F^0(V)}{F^1(V)}. \tag{12}$$

$R_V$ is a Poisson algebra [66, 71] and is finitely generated if and only if $V$ is strongly finitely generated. Moreover the image of generators of $V$ in $R_V$ generates $R_V$ as well. Notice that $R_V$ is in general not reduced, namely the ideal defining it would contain a nilpotent element [7]. The product and Poisson structure on $R_V$ are defined as

$$\bar{a} \cdot \bar{b} = \overline{a_{-1}b}, \quad \{\bar{a}, \bar{b}\} = \overline{a_0 b}. \tag{13}$$

We have now an associated scheme and an associated variety defined from Zhu's $C_2$ algebra

$$\tilde{X}_V = \mathrm{spec}(R_V), \qquad X_V = \mathrm{spec}((R_V)_{red}). \tag{14}$$

$X_V$ is a Poisson variety [71, 73]. If $X_V$ is a smooth variety, one may view $X_V$ as a complex-analytic manifold equipped with a holomorphic Poisson structure, and for each point $x \in X_V$, there is a well-defined symplectic $S_x$ leaf through $x$, which is the set of points that can be reached from $x$ by going along Hamiltonian flows. If $X_V$ is not necessarily smooth, let $Sing(X_V)$ be the singular locus of $X$, and for any $k \geq 1$ define inductively $Sing^k(X_V) := Sing(Sing^{k-1}(X_V))$. We get a finite partition

$$X_V = \cup_k X_V^k, \tag{15}$$

where the strata $X_V^k := Sing^{k-1}(X_V)Sing^k(X_V)$ are smooth analytic varieties (more details can be found in [74–76]). It is known that each $X_V^k$ inherits a Poisson structure [73]. So for any point of $x \in X_V^k$ there is a well-defined symplectic leaf $S_x \subset X_V^k$. In this way one defines symplectic leaves on an arbitrary Poisson variety.

A **lisse** VOA is defined as the VOA such that $\dim(X_V) = 0$ (see for example [71, 77, 78]). A rational VOA has to be lisse, but it is an open problem to prove that lisse VOA has to be rational. A quasi-lisse VOA is defined as the VOA whose associated variety $X_V$ has finite number of symplectic leaves. Quasi-lisse VOA has many interesting properties [31, 79]:

---

[7]A nilpotent element $x$ of an ideal is an element not in $I$ but $x^n \in I$ for some $n$.

- The VOA is strongly finitely generated.

- The Virasoro vector $\omega_V$ is nilpotent in $R_V$.

- There are finite number of ordinary modules, and they transform nicely under modular transformations. A weak $V$-module $(M, Y_M)$ is called *ordinary* if $L_0$ acts semi-simply on $M$, any $L_0$-eigenspace $M_\Delta$ of $M$ of eigenvalue $\Delta \in \mathbb{C}$ is finite-dimensional, and for any $\Delta \in \mathbb{C}$, $M_{\Delta-n} = 0$ for all sufficiently large $n \in \mathbb{Z}$.

- The character satisfies a modular differential equation.

## 2.2 4d/2d correspondence

It was proposed in [5] that one can get a 2d VOA from the Schur sector of a 4d $\mathcal{N} = 2$ SCFT, and the basic 4d/2d dictionary used in current paper is [5]:

- There is an AKM subalgebra ($V^{k_{2d}}(\mathfrak{g})$) in 2d VOA, where $\mathfrak{g}$ is the Lie algebra of 4$d$ flavor symmetry $G_F$.

- The 2d central charge $c_{2d}$ and the level of AKM algebra $k_{2d}$ are related to the 4d central charge $c_{4d}$ and the flavor central charge $k_F$ as

$$c_{2d} = -12c_{4d}, \quad k_{2d} = -k_F{}^8. \tag{16}$$

- The (normalized) vacuum character of 2d VOA is the 4d Schur index $\mathcal{I}(q)$.

- The associated variety is the Higgs branch of the 4d $\mathcal{N} = 2$ SCFT [29, 31, 67].

## 2.3 Comments on constraints of 2d VOAs corresponding to 4d SCFTs

It is conjectured that the VOA corresponding to a 4d $\mathcal{N} = 2$ SCFT is always a quasi-lisse VOA [31]. However, not all lisse VOA has a 4d $\mathcal{N} = 2$ SCFT counterpart. We do have some constraints based on 4d unitarity:

- The 2d central charge is negative and has to satisfy the constraint $c_{2d} \leq -\frac{11}{30}$ for interacting 4d $\mathcal{N} = 2$ SCFTs [80].

- If 4d theory has a flavor group $G$, its level is bounded from below $k_G \geq k_{critical}$ [5], so the corresponding 2d AKM level is also constrained.

- The minimal conformal weight of primary fields of VOA is constrained to be $\frac{c_{2d}}{8} \leq h_{min} \leq 0$ [31].

These constraints come from considerations purely on the Schur sector. On the other hand, there are some very mysterious relations between the Schur sector and the Coulomb branch data:

1. First, one can compute the central charge $a_{4d}$ and $c_{4d}$ purely from Coulomb branch data. $c_{4d}$ is obviously related to the 2d VOA, and $a_{4d} - c_{4d}$ is also related to the asymptotic limit of the Schur index, see [31] and further discussions in section 4.5.

2. One can compute the Schur index from the Coulomb branch massive BPS spectrum [9, 13].

---

[8]Our normalization of $k_F$ is half of that of [5, 6].

3. If we know the common denominator $r$ of Coulomb branch operators, the flavor central charge seem to be bounded by a number which depends on the denominator $r$ [81]. This bound is different from the minimal bound found from Higgs branch data only.

So from this perspective, the bound from purely Higgs branch data seems to be not enough on constraining the set of quasi-lisse VOA which can be VOA of 4d theory. With input from Coulomb branch data, one can get much stronger constraint on VOA, and we plan to study this further in the near future.

# 3 Argyres-Douglas theories and $W$-algebras

In this section we review known results on the classification of AD theories from $M5$ branes and their corresponding VOAs. We focus on AD theories whose VOAs are $W$-algebras at boundary admissible levels.

## 3.1 AD theories correspond to $W^{k'}(\mathfrak{g}, f)$ algebras

One can engineer a large class of 4d $\mathcal{N} = 2$ SCFTs by starting from a 6d $(2,0)$ theory of type $\mathfrak{j} = ADE$ on a sphere with an irregular singularity and a regular singularity [39,58,59,82,83]. The Coulomb branch is captured by a Hitchin system with singular boundary conditions near the singularity. The Higgs field of the Hitchin system near the irregular singularity takes the following form

$$\Phi = \frac{T}{z^{2+\frac{k}{b}}} + \dots, \tag{17}$$

where $T$ is determined by a positive grading of Lie algebra $\mathfrak{j}$ [84], and is a regular semi-simple element of $\mathfrak{j}$. $k$ is an integer greater than $b$. Subsequent terms are chosen such that they are compatible with the leading order term (essentially the grading determines the choice of these terms). We call them $J^{(b)}[k]$ type irregular puncture. Theories constructed using only above irregular singularities can also be engineered using a three dimensional singularity in type IIB string theory as summarized in table 1 [85].

Table 1: Three-fold isolated quasi-homogenous singularities of cDV type corresponding to the $J^{(b)}[k]$ irregular punctures of the regular-semisimple type in [59]. These 3d singularity is very useful in extracting the Coulomb branch spectrum [85].

| $\mathfrak{j}$ | $b$ | Singularity |
|---|---|---|
| $A_{N-1}$ | $N$ | $x_1^2 + x_2^2 + x_3^N + z^k = 0$ |
| | $N-1$ | $x_1^2 + x_2^2 + x_3^N + x_3 z^k = 0$ |
| $D_N$ | $2N-2$ | $x_1^2 + x_2^{N-1} + x_2 x_3^2 + z^k = 0$ |
| | $N$ | $x_1^2 + x_2^{N-1} + x_2 x_3^2 + z^k x_3 = 0$ |
| $E_6$ | $12$ | $x_1^2 + x_2^3 + x_3^4 + z^k = 0$ |
| | $9$ | $x_1^2 + x_2^3 + x_3^4 + z^k x_3 = 0$ |
| | $8$ | $x_1^2 + x_2^3 + x_3^4 + z^k x_2 = 0$ |
| $E_7$ | $18$ | $x_1^2 + x_2^3 + x_2 x_3^3 + z^k = 0$ |
| | $14$ | $x_1^2 + x_2^3 + x_2 x_3^3 + z^k x_3 = 0$ |
| $E_8$ | $30$ | $x_1^2 + x_2^3 + x_3^5 + z^k = 0$ |
| | $24$ | $x_1^2 + x_2^3 + x_3^5 + z^k x_3 = 0$ |
| | $20$ | $x_1^2 + x_2^3 + x_3^5 + z^k x_2 = 0$ |

One can add another regular singularity which is labeled by a nilpotent orbit $f$ of $\mathfrak{j}$ (We use Nahm labels such that the trivial orbit corresponding to a regular puncture with maximal flavor symmetry). A detailed discussion on these defects can be found in [86].

To get non-simply laced flavor groups, we need to consider the outer-automorphism twist of ADE Lie algebra and its Langlands dual. A systematic study of these AD theories was performed in [39]. Denoting the twisted Lie algebra of $\mathfrak{j}$ as $\mathfrak{g}^\vee$ and its Langlands dual as $\mathfrak{g}$, outer-automorphisms and twisted algebras of $\mathfrak{j}$ are summarized in table 2. The irregular singularity of regular semi-simple type is also classified in table 3 with the following form

$$\Phi = \frac{T^t}{z^{2+\frac{k}{b}}} + \dots \tag{18}$$

Here $T^t$ is an element of Lie algebra $\mathfrak{g}^\vee$ or other parts of the decomposition of $\mathfrak{j}$ under outer automorphism. $k > -b$, and the novel thing is that $k$ could take half-integer value or in thirds ($\mathfrak{g} = G_2$). One can also represent those irregular singularities by 3-fold singularities as in table 3.

Table 2: Outer-automorphisms of simple Lie algebras $\mathfrak{j}$, its invariant subalgebra $g^\vee$ and flavor symmetry $g$ from the Langlands dual of $g^\vee$.

| $\mathfrak{j}$ | $A_{2N}$ | $A_{2N-1}$ | $D_{N+1}$ | $E_6$ | $D_4$ |
|---|---|---|---|---|---|
| Outer-automorphism $o$ | $Z_2$ | $Z_2$ | $Z_2$ | $Z_2$ | $Z_3$ |
| Invariant subalgebra $\mathfrak{g}^\vee$ | $B_N$ | $C_N$ | $B_N$ | $F_4$ | $G_2$ |
| Flavor symmetry $\mathfrak{g}$ | $C_N^{(1)}$ | $B_N$ | $C_N^{(2)}$ | $F_4$ | $G_2$ |

Table 3: Seiberg-Witten geometry of twisted theories at the SCFT point.

| $\mathfrak{j}$ with twist | $b_t$ | SW geometry at SCFT point | $\Delta[z]$ |
|---|---|---|---|
| $A_{2N}/Z_2$ | $4N+2$ | $x_1^2 + x_2^2 + x^{2N+1} + z^{k+\frac{1}{2}} = 0$ | $\frac{4N+2}{4N+2k+3}$ |
| | $2N$ | $x_1^2 + x_2^2 + x^{2N+1} + xz^k = 0$ | $\frac{2N}{k+2N}$ |
| $A_{2N-1}/Z_2$ | $4N-2$ | $x_1^2 + x_2^2 + x^{2N} + xz^{k+\frac{1}{2}} = 0$ | $\frac{4N-2}{4N+2k-1}$ |
| | $2N$ | $x_1^2 + x_2^2 + x^{2N} + z^k = 0$ | $\frac{2N}{2N+k}$ |
| $D_{N+1}/Z_2$ | $2N+2$ | $x_1^2 + x_2^N + x_2 x_3^2 + x_3 z^{k+\frac{1}{2}} = 0$ | $\frac{2N+2}{2k+2N+3}$ |
| | $2N$ | $x_1^2 + x_2^N + x_2 x_3^2 + z^k = 0$ | $\frac{2N}{k+2N}$ |
| $D_4/Z_3$ | $12$ | $x_1^2 + x_2^3 + x_2 x_3^2 + x_3 z^{k\pm\frac{1}{3}} = 0$ | $\frac{12}{12+3k\pm1}$ |
| | $6$ | $x_1^2 + x_2^3 + x_2 x_3^2 + z^k = 0$ | $\frac{6}{6+k}$ |
| $E_6/Z_2$ | $18$ | $x_1^2 + x_2^3 + x_3^4 + x_3 z^{k+\frac{1}{2}} = 0$ | $\frac{18}{18+2k+1}$ |
| | $12$ | $x_1^2 + x_2^3 + x_3^4 + z^k = 0$ | $\frac{12}{12+k}$ |
| | $8$ | $x_1^2 + x_2^3 + x_3^4 + x_2 z^k = 0$ | $\frac{8}{12+k}$ |

We could again add a twisted regular puncture labeled also by a nilpotent orbit $f$ of $\mathfrak{g}$. If there is no mass parameter in the irregular singularity, the corresponding VOA is given by the following $W$ algebra [39]

$$\boxed{W^{k'}(\mathfrak{g}, f), \quad k' = -h^\vee + \frac{1}{n}\frac{b}{k+b},} \tag{19}$$

where $h^\vee$ is the dual Coxeter number of $\mathfrak{g}$, $n$ is the number listed in table 4, and $k$ is restricted to the value such that no mass parameter is in the irregular singularity.

In this paper, we are going to focus on the choice of $b$ and $n$ such that the corresponding $W$ algebra takes the following form

$$\boxed{W^{k'}(\mathfrak{g}, f), \quad k' = -h^\vee + \frac{h^\vee}{k + h^\vee}}, \quad (k, h^\vee) = 1. \tag{20}$$

There are some further constraints on value $k$: a) $h^\vee + k \geq 2$; b) $k$ and $h^\vee$ coprime; and c) $k \neq 2n$ for $\mathfrak{g} = B_N, C_N, F_4$, and $k \neq 3n$ for $\mathfrak{g} = G_2$.

Table 4: Lie algebra data. $h$ is the Coxeter number and $h^\vee$ is the dual Coxeter number.

|  | dimension | $h$ | $h^\vee$ | $n$ |
|---|---|---|---|---|
| $A_{N-1}$ | $N^2 - 1$ | $N$ | $N$ | 1 |
| $B_N$ | $(2N+1)N$ | $2N$ | $2N - 1$ | 2 |
| $C_N^{(1)}$ | $(2N+1)N$ | $2N$ | $N + 1$ | 4 |
| $C_N^{(2)}$ | $(2N+1)N$ | $2N$ | $N + 1$ | 2 |
| $D_N$ | $N(2N-1)$ | $2N - 2$ | $2N - 2$ | 1 |
| $E_6$ | 78 | 12 | 12 | 1 |
| $E_7$ | 133 | 18 | 18 | 1 |
| $E_8$ | 248 | 30 | 30 | 1 |
| $F_4$ | 52 | 12 | 9 | 2 |
| $G_2$ | 14 | 6 | 4 | 3 |

# 4 The character of $W$-algebra and the Schur index

Now we discuss Schur indices of AD theories from their corresponding $W$-algebras. The index can be written in a simplified form which implies many interesting properties of the SCFT and the VOA.

## 4.1 The $W$-algebra from the qDS reduction

We first set up the notation for Lie algebra datas. Let $\mathfrak{g}$ be a simple finite dimensional Lie algebra, and let $\mathfrak{h}$ be the Cartan subalgebra of $\mathfrak{g}$, and let $\Delta \subset \mathfrak{h}^*$ be the set of roots, where $\mathfrak{h}^*$ is the dual space of $\mathfrak{h}$. Let $Q = \mathbb{Z}\Delta$ be the root lattice and let $Q^* = \{h \in \mathfrak{h} | \alpha(h) \in \mathbb{Z} \text{ for all } \alpha \in \Delta\}$ be its dual lattice. We also use $\Delta_+$ to denote the set of positive roots, and $\{\alpha_1, \ldots, \alpha_l\}$ be the set of simple roots with $l$ be the rank of $\mathfrak{g}$. We denote $\rho$ as the half of the sum of all positive roots. The bracket $(\cdot|\cdot)$ is the invariant bilinear form on $\mathfrak{g}$ with the normalization $(\alpha|\alpha) = 2$ for the long roots. $h^\vee$ is the dual Coxeter number. We use $\omega_i$ to denote the fundamental weights of Lie algebra $\mathfrak{g}$.

Now for AKM algebra $\hat{\mathfrak{g}} = \mathfrak{g}[t, t^{-1}] + CK + Cd$, its Cartan subalgebra is $\hat{\mathfrak{h}} = \mathfrak{h} + CK + Cd$. The bilinear form on AKM algebra is extended from the bilinear form of $\mathfrak{g}$ as follows

$$(\mathfrak{h}|CK + Cd) = 0, \quad (K|K) = 0, \quad (d|d) = 0, \quad (d|K) = 1. \tag{21}$$

We can use this bilinear form to identify the dual space $\hat{h}^*$ with $\hat{h}$. Roots of AKM are denoted by three sets of eigenvalues. The imaginary root has the label $\delta = (0, 0, 1)$ and simple roots are $\hat{\alpha}_i = (\alpha_i, 0, 0)$ with $\alpha_i$ being simple roots of $\mathfrak{g}$. Furthermore we have the zeroth simple root $\hat{\alpha}_0 = (-\theta, 0, 1)$ with $\theta$ being the highest root of $\mathfrak{g}$. The set of real roots are

$\hat{\Delta}^{re} = \{\alpha + n\delta | \alpha \in \Delta, n \in \mathbb{Z}\}$, and the set of positive real roots is denoted as $\hat{\Delta}^{re}_+ = \Delta_+ \cup \{\alpha + n\delta | \alpha \in \Delta, n \in \mathbb{Z}_{\geq 1}\}$. Affine fundamental weights are $\Lambda_0 = (0, 1, 0)$ and $\Lambda_i = (\omega_i, a_i^\vee, 0)$ with $a_i^\vee$ being the comark which is 1 for simply laced Lie algebra. We also define $\hat{\rho} = \sum_{i=0}^{l} \hat{\omega}_i$. One has following important set of roots

$$\hat{\Pi}_u = \{u\delta - \theta, \hat{\alpha}_1, \ldots, \hat{\alpha}_l\}, \tag{22}$$

which is used in defining principal admissible weights.

For a $\beta \in Q^*$, one define a translation $t_\beta \in End(\hat{h}^*)$ with the following formula

$$t_\beta(\lambda) = \lambda + \lambda(K)\beta - ((\lambda, \beta) + \frac{1}{2}\lambda(K)|\beta|^2)\delta. \tag{23}$$

An element in the extended affine Weyl group $\tilde{W}$ can be written in the form $t_\beta y$ with $y \in W$ an element in Weyl group of lie algebra $\mathfrak{g}$.

Now $\Lambda$ is called a principal admissible weight if the following two properties hold

1. The level $k = \Lambda(K)$ is a rational number with denominator $u \in \mathbb{Z}_{\geq 1}$, such that

$$k + h^\vee \geq \frac{h^\vee}{u} \text{ and } \gcd(u, h^\vee) = \gcd(u, r^\vee) = 1, \tag{24}$$

   where $r^\vee$ takes 1 for $\mathfrak{g}$ of type ADE, and 2 for $\mathfrak{g}$ of type B, C, F, and 3 for $\mathfrak{g} = G_2$.

2. All principal admissible weights $\Lambda$ are of the form

$$\Lambda = (t_\beta y).(\Lambda^0 - (u-1)(k + h^\vee)\Lambda_0), \tag{25}$$

   where $\beta \in Q^*$, $y \in W$ are such that $(t_\beta y)\hat{\Pi}_u \subset \hat{\Delta}_+$, $\Lambda^0$ is an integrable weight of level $u(k + h^\vee) - h^\vee$, and dot denotes the shifted action $w.\Lambda = w(\Lambda + \hat{\rho}) - \hat{\rho}$.

Starting with an AKM algebra $V^k(\mathfrak{g})$, one can get a large class of $W$ algebras by using the quantum Drinfeld-Soklov reduction [61]. Given a $\mathfrak{sl}_2$ triple $(x, e, f)$ with the nilpotent element $f$, and the commutation relation is defined as

$$[x, e] = e, \ [x, f] = -f, \ [e, f] = 2x. \tag{26}$$

The corresponding $W$ algebra is denoted as $W^k(\mathfrak{g}, f)$. The universal $W$ algebra has following properties: it is finitely strongly generated by the following fields $J_{v_j}$ with scaling dimension $1 - j$. Here $v_j \in \mathfrak{g}_j^f$ with $j \leq 0$. Let's explain the notation now: Given a $\mathfrak{sl}_2$ triple $(x, e, f)$ with $x$ a semi-simple element, we can decompose $\mathfrak{g}$ as: $\mathfrak{g} = \oplus \mathfrak{g}_j$ with $\mathfrak{g}_j = \{[x, g_j] = jg_j\}$. $\mathfrak{g}_j^f$ is defined as the elements in $\mathfrak{g}_j$ which also commutes with nilpotent element $f$. There is a symmetry between $\pm j$ such that $\dim \mathfrak{g}_j^f = \dim \mathfrak{g}_{-j}^f$.

## 4.2 Character of $W$-algebra modules

For admissible modules of AKM and corresponding W-algebras at boundary level, their characters decompose in products in terms of the Jacobi form $\theta_{11}(\tau, z)$ [63]. This result provides an elegant closed form formula for Schur indices of the AD theory discussed in this paper.

Starting with AKM at boundary level $k = -h^\vee + \frac{h^\vee}{u}$, all boundary principal admissible weights are of the form

$$\Lambda = (t_\beta y).(k\Lambda_0), \tag{27}$$

where $\beta \in Q^*$, $y \in W$ are such that $(t_\beta y)\hat{\Pi}_u \subset \hat{\Delta}_+$. The character of the module corresponding to admissible weight $\Lambda$ can be expressed in products of theta functions [63]

$$\text{ch}_\Lambda(\tau, z, t) = e^{2\pi i\left(kt + \frac{h^\vee}{u}(z|\beta)\right)} q^{\frac{h^\vee}{2u}|\beta|^2} \left(\frac{\eta(u\tau)}{\eta(\tau)}\right)^{\frac{1}{2}(3l-\dim\mathfrak{g})} \prod_{\alpha\in\Delta_+} \frac{\theta_{11}(y(\alpha)(z+\tau\beta), u\tau)}{\theta_{11}(\alpha(z), \tau)}. \quad (28)$$

Convention of $\eta(\tau)$ and $\theta_{11}$ are summarized in appendix A. In particular the vacuum module has the weight $k\Lambda_0$, and its character is

$$\text{ch}_{k\Lambda_0}(\tau, z, t) = e^{2\pi ikt} \left(\frac{\eta(u\tau)}{\eta(\tau)}\right)^{\frac{1}{2}(3l-\dim\mathfrak{g})} \prod_{\alpha\in\Delta_+} \frac{\theta_{11}(\alpha(z), u\tau)}{\theta_{11}(\alpha(z), \tau)}. \quad (29)$$

The Schur index of the corresponding AD theory is obtained simply by setting $t = 0$ and normalizing the character such that the Schur index goes to one when $q = e^{2\pi i\tau}$ goes to zero.

For W-algebra $W^k(\mathfrak{g}, f)$ from the vacuum $\hat{\mathfrak{g}}$-module of level $k$ by the qDS reduction, there is a reductive functor $H$ which maps principal admissible modules of AKM to either zero or an irreducible module of $W^k(\mathfrak{g}, f)$. The character of the irreducible $W^k(\mathfrak{g}, f)$-module $H(\Lambda)$ is

$$\begin{aligned}
\text{ch}_{H(\Lambda)}(\tau, z) =&(-i)^{|\Delta_+|} q^{\frac{h^\vee}{2u}|\beta-x|^2} e^{\frac{2\pi i h^\vee}{u}(\beta|z)} \\
&\times \frac{\eta(u\tau)^{\frac{3}{2}l - \frac{1}{2}\dim\mathfrak{g}}}{\eta(\tau)^{\frac{3}{2}l - \frac{1}{2}\dim(\mathfrak{g}_0 + \mathfrak{g}_{\frac{1}{2}})}} \frac{\prod_{\alpha\in\Delta_+} \theta_{11}(y(\alpha)(z+\tau\beta-\tau x), u\tau)}{\prod_{\alpha\in\Delta_+^0} \theta_{11}(\alpha(z), \tau)\left(\prod_{\alpha\in\Delta_{\frac{1}{2}}} \theta_{01}(\alpha(z), \tau)\right)^{\frac{1}{2}}},
\end{aligned} \quad (30)$$

where $\{f, x, e\}$ forms the $\mathfrak{sl}_2$-triple in $\mathfrak{g}$, $\mathfrak{g} = \oplus_j \mathfrak{g}_j$ is the eigenspace decomposition for ad$x$, $\Delta_j \subset \Delta$ is the set of roots of the root spaces in $\mathfrak{g}_j$ and $\Delta_+^0 = \Delta_+ \cap \Delta_0$. If the reduction of $\Lambda$ gives zero, $\text{ch}_{H(\Lambda)} = 0$ automatically. If $\Lambda_1$ and $\Lambda_2$ lead to the same module in the W-algebra, $\text{ch}_{H(\Lambda_1)} = \text{ch}_{H(\Lambda_2)}$. In particular the vacuum module of the W-algebra is $H(k\Lambda_0)$ with the character

$$\text{ch}_{H(k\Lambda_0)}(\tau, z) = (-i)^{|\Delta_+|} q^{\frac{h^\vee}{2u}|x|^2} \frac{\eta(u\tau)^{\frac{3}{2}l - \frac{1}{2}\dim\mathfrak{g}}}{\eta(\tau)^{\frac{3}{2}l - \frac{1}{2}\dim(\mathfrak{g}_0 + \mathfrak{g}_{\frac{1}{2}})}} \frac{\prod_{\alpha\in\Delta_+} \theta_{11}(\alpha(z-\tau x), u\tau)}{\prod_{\alpha\in\Delta_+^0} \theta_{11}(\alpha(z), \tau)\left(\prod_{\alpha\in\Delta_{\frac{1}{2}}} \theta_{01}(\alpha(z), \tau)\right)^{\frac{1}{2}}}. \quad (31)$$

It also gives the Schur index of the corresponding AD theory after normalization.

## 4.3 The simplified form

Using the product formula in the previous section, we can put the index of $W^k(\mathfrak{g}, f)$ in a even simpler form. If $f$ is regular principal, the index is thus

$$\mathcal{I}_{W^{k'}(\mathfrak{g}, f_{prin})} = PE\left[\frac{\sum_i q^{d_i} - q^{h^\vee + k + 1}(\sum q^{-d_i})}{(1-q)(1-q^{h^\vee + k})}\right], \quad k' = -h^\vee + \frac{h^\vee}{h^\vee + k}, \quad (32)$$

where $h^\vee$ is the dual Coxeter number, the plethystic exponential $PE$ is defined as

$$PE[f(a, \cdots)] = \exp\left[\sum_{n=1}^\infty \frac{1}{n} f(a^n, \cdots)\right], \quad (33)$$

and $\{d_i\}$ is the set of degrees of Casimiars of Lie algebra $\mathfrak{g}$. For example, degrees of Casimir of $A_N$ Lie algebra are $\{2, 3, \cdots, N+1\}$. On the other hand, if $f$ is trivial, the index becomes

$$\mathcal{I}_{W^{k'}(\mathfrak{g}, f_{tri})} = PE\left[\frac{q - q^{h^\vee + k}}{(1-q)(1-q^{h^\vee + k})} \chi_{adj}(z)\right], \quad k' = -h^\vee + \frac{h^\vee}{h^\vee + k}, \quad (34)$$

where $\chi_{adj}(z)$ is the character of the adjoint representation of $\mathfrak{g}$. For generic $f$, the Lie group $G$ of $\mathfrak{g}$ has a subgroup $SU(2) \times G_F$ with $G_F$ being the flavor group determined by $f$. Under this subgroup the adjoint representation of $G$ decomposes as

$$adj_G = \sum_j V_j \otimes R_j, \tag{35}$$

where $V_j$ is the spin $j$ representation of $SU(2)$ and $R_j$ is the corresponding representation of $G_F$. The Schur index takes the following form ($k' = -h^\vee + \frac{h^\vee}{h^\vee + k}$)

$$\mathcal{I}_{W^{k'}(\mathfrak{g},f)} = PE\left[\sum \frac{q^{1+j}}{1-q}\chi_{R_j}(z) - \frac{q^{h^\vee+k}}{1-q^{h^\vee+k}}\sum_j \chi_{V_j}(q)\chi_{R_j}(z)\right], \tag{36}$$

with $\chi_{V_j}$ being the character of spin $j$ representation of $SU(2)$ and $\chi_{R_j}(z)$ being the character of $R_j$ defined as $\mathrm{tr}_{R_j} e^{2\pi i z}$ with $z \in \mathfrak{h}^f$ and $\mathfrak{h}^f$ are the Cartan of $\mathfrak{g}_0^f$ (centralizer of the $\mathfrak{sl}_2$ triple $(x, e, f)$). The dimension of $R_j$ is the same as the dimension of $\mathfrak{g}_{\pm j}^f$. We rewrite the character as follows

$$\mathcal{I}_{W^{k'}(\mathfrak{g},f)} = PE\left[\frac{\sum_j q^{1+j}\chi_{R_j}(z)}{(1-q)(1-q^{h^\vee+k})} - \frac{q^{h^\vee+k}}{(1-q)(1-q^{h^\vee+k})}\left[\sum q^{1+j}\chi_{R_j}(z) + (1-q)\sum_j \chi_{V_j}(q)\chi_{R_j}(z)\right]\right]. \tag{37}$$

To further simplify this expression, we use the following identity

$$\chi_{V_j}(q) = \frac{q^{-j-1/2} - q^{j+1/2}}{q^{-1/2} - q^{1/2}} = \frac{q^{-j} - q^{j+1}}{1-q}, \tag{38}$$

and finally our index for generic $f$ takes the following form

$$\mathcal{I}_{W^{k'}(g,f)} = PE\left[\frac{\sum_j q^{1+j}\chi_{R_j}(z) - q^{h^\vee+k}\sum_j q^{-j}\chi_{R_j}(z)}{(1-q)(1-q^{h^\vee+k})}\right]. \tag{39}$$

### 4.3.1 Applications

**Level-Rank duality:** One can check the level-rank duality using the index formula 32. For example, the Schur index of $(A_{N-1}, A_{k-1})$ AD theory is

$$\mathcal{I}_{(A_{N-1},A_{k-1})} = PE\left[\frac{(1-q^{k-1})\sum_{j=2}^N q^j}{(1-q)(1-q^{N+k})}\right] = PE\left[\frac{q^2(1-q^{k-1})(1-q^{N-1})}{(1-q)^2(1-q^{N+k})}\right], \tag{40}$$

which is symmetric under the exchange of $k$ and $N$, reproducing the result in [29]. For $(G_1, G_2)$ theories with $\gcd(h_1^\vee, h_2^\vee) = 1$, the Schur index can also be written as

$$\mathcal{I}_{(G_1,G_2)} = PE\left[\frac{\left(\sum_i q^{d_i^{(1)}}\right)\left(\sum_j q^{d_j^{(2)}}\right)}{q^2(1-q^{h_1^\vee+h_2^\vee})}\right], \tag{41}$$

where $d_i^{(r)}$'s are degrees of Casimirs of $G_r$.

This type of level rank duality is vastly generalized in [57]. One example is the following identification of $W$-algebras

$$
W^{-n1(n+k)-n+\frac{n_1(n+k)+n}{n+k}}(\mathfrak{sl}_{n_1(n+k)+n}, [(n+k-1)^{n_1}, n+n_1])
$$
$$
= W^{-k+\frac{k}{n+k}}(\mathfrak{sl}_k, [k-n_1, 1^{n_1}]). \tag{42}
$$

To compute the Schur index of the RHS, notice that the character of the adjoint representation of $\mathfrak{sl}_k$ decomposes under $[k-n_1, 1^{n_1}]$ as

$$
\chi^{\mathfrak{sl}_k}_{adj} = \chi_{adj} + (a^{2n_1-k}\chi_{\square} + a^{-2n_1+k}\chi_{\overline{\square}})\chi_{V_{\frac{k-n_1-1}{2}}} + \sum_{j=0}^{k-n_1-1} \chi_{V_j}, \tag{43}
$$

where $\chi_{adj}$, $\chi_{\square}$ and $\chi_{\overline{\square}}$ are the characters of adjoint, fundamental and anti-fundamental representations of $SU(n_1)$ respectively. Therefore the Schur index for the RHS follows the simplified formula [39]

$$
\mathcal{I}_{RHS} = PE\left[\frac{f_{RHS}}{(1-q)(1-q^{n+k})}\right], \tag{44}
$$

where $f_{RHS}$ is

$$
f_{RHS} = (q-q^{n+k})\chi_{adj} + (q^{\frac{k-n_1+1}{2}} - q^{n+\frac{k+n_1+1}{2}})(\chi_{\square}a^{2n_1-k} + \chi_{\overline{\square}}a^{k-2n_1}) + \sum_{j=0}^{k-n_1-1}(q^{1+j} - q^{n+k-j})
$$
$$
= (q-q^{n+k})\chi_{adj} + (q^{\frac{k-n_1+1}{2}} - q^{n+\frac{k+n_1+1}{2}})(\chi_{\square}a^{2n_1-k} + \chi_{\overline{\square}}a^{k-2n_1}) + \frac{(q-q^{k-n_1+1})(1-q^{n+n_1})}{1-q}. \tag{45}
$$

On the other hand, the character of the adjoint representations of $\mathfrak{sl}_{n_1(n+k)+n}$ decomposes under $[(n+k-1)^{n_1}, n+n_1]$ as

$$
\chi^{\mathfrak{sl}_{n_1(n+k)+n}} = (\chi_{adj}+1)\sum_{j=0}^{n+k-2}\chi_{V_j} + \sum_{j=1}^{n+n_1-1}\chi_{V_j}
$$
$$
+ (b^{n-n_1(n+k)}\chi_{\square} + b^{n_1(n+k)-n}\chi_{\overline{\square}})\sum_{j=(k-n_1-1)/2}^{n+(n_1+k-3)/2}\chi_{V_j}, \tag{46}
$$

hence the Schur index for the LHS is

$$
\mathcal{I}_{LHS} = PE\left[\frac{f_{LHS}}{(1-q)(1-q^{n+k})}\right], \tag{47}
$$

with

$$
f_{LHS} = \left(\sum_{j=0}^{n+k-2} q^{1+j} - q^{n+k-j}\right)(\chi_{adj}+1) + \sum_{j=1}^{n+n_1-1}(q^{1+j} - q^{n+k-j})
$$
$$
+ (b^{n-n_1(n+k)}\chi_{\square} + b^{n_1(n+k)-n}\chi_{\overline{\square}})\sum_{j=(k-n_1-1)/2}^{n+(n_1+k-3)/2}(q^{1+j} - q^{n+k-j})
$$
$$
= (q-q^{n+k})\chi_{adj} + \frac{(q-q^{k-n_1+1})(1-q^{n+n_1})}{1-q}
$$
$$
+ (q^{\frac{k-n_1+1}{2}} - q^{n+\frac{k+n_1+1}{2}})(b^{n-n_1(n+k)}\chi_{\square} + b^{n_1(n+k)-n}\chi_{\overline{\square}}). \tag{48}
$$

Comparing equation 45 and 48, we see $f_{LHS} = f_{RHS}$ with a redefinition of $b$, therefore $\mathcal{I}_{LHS} = \mathcal{I}_{RHS}$, providing another check of the generalized level-rank duality 42.

**Collapsing levels**: One may also understand the phenomenon of collapsing levels [57, 87] by using the character formula 39. For example, consider $\mathfrak{g} = \mathfrak{sl}_N$ and the nilpotent orbit labelled by the Young tableaux $Y = [2, 1^{N-2}]$. The flavor group is then $SU(N-2) \times U(1)$. The character of the adjoint representation of $\mathfrak{sl}_N$ decomposes under such nilpotent element as

$$\chi_{adj}^{\mathfrak{sl}_N} = (\chi_{adj} + 1) + \left(a\chi_{\square} + \frac{1}{a}\chi_{\square}\right)\chi_{V_{1/2}} + \chi_{V_1}, \tag{49}$$

where the fugacity $a$ is the exponential of the $U(1)$ Cartan and $\chi_{adj}$, $\chi_{\square}$ and $\chi_{\square}$ are the character of adjoint, fundamental and anti-fundamental representation of $SU(N-2)$ respectively. One sees immediately from the formula 39 that terms proportional to $\chi_{V_{1/2}}$ cancel with each other when $k + h^\vee = 2$

$$PE\left[\frac{q(\chi_{adj}+1)+q^2-q^2(\chi_{adj}+1)-q}{(1-q)(1-q^2)}\right] = PE\left[\frac{q\chi_{adj}}{1-q^2}\right], \tag{50}$$

hence the character reduces to the character of the affine $\mathfrak{sl}_{N-2}$ algebra with the level $\tilde{k} = -(N-2) + \frac{N-2}{2}$.

For $\mathfrak{g} = \mathfrak{sl}_N$ and $Y = [r^m, 1^{N-rm}]$, the flavor symmetry is $G_F = SU(m) \times SU(N-rm) \times U(1)$. The character of the adjoint representation of $\mathfrak{sl}_N$ decomposes under $SU(2) \times G_F$ as

$$\begin{aligned}\chi_{adj}^{\mathfrak{sl}_N} = &\chi_{adj}^{SU(N-rm)} + (\chi_{V_0} + \chi_{V_1} + \cdots + \chi_{V_{r-1}})(\chi_{adj}^{SU(m)} + 1) \\ &+ (a^{N-rm-m}\chi_{\square}^{SU(m)}\chi_{\overline{\square}}^{SU(N-rm)} + a^{m-N+rm}\chi_{\overline{\square}}^{SU(m)}\chi_{\square}^{SU(N-rm)})\chi_{V_{\frac{r-1}{2}}},\end{aligned} \tag{51}$$

again the fugacity $a$ labels the $U(1)$ symmetry. At $k + h^\vee = r$, the generator and relation which contain both representations of $SU(m)$ and $SU(N-rm)$ cancel with each other and the character is

$$\begin{aligned}PE&\left[\frac{(q-q^r)\chi_{adj}^{SU(N-rm)}}{(1-q)(1-q^r)} + \frac{(q-q^r)+(q^2-q^{r-1})+\cdots+(q^r-q)}{(1-q)(1-q^r)}(\chi_{adj}^{SU(m)}+1)\right] \\ &= PE\left[\frac{(q-q^r)\chi_{adj}^{SU(N-rm)}}{(1-q)(1-q^r)}\right],\end{aligned} \tag{52}$$

which is the same as the vacuum character of the affine $\mathfrak{sl}_{N-rm}$ with level $\tilde{k} = -h^\vee + \frac{h^\vee}{r}$.

For $\mathfrak{g} = \mathfrak{so}_{2N}$ and $Y = [r^m, 1^{2N-rm}]$ with $r$ being an odd number, the flavor symmetry is $G_F = SO(m) \times SO(2N-rm)$. The character of the adjoint representation of $\mathfrak{so}_{2N}$ decomposes under $SU(2) \times G_F$ as

$$\begin{aligned}\chi_{adj}^{\mathfrak{so}_{2N}} = &\chi_{adj}^{SO(2N-rm)} + (\chi_{V_0} + \chi_{V_2} + \cdots + \chi_{V_{r-1}})\chi_{asym^2\square}^{SO(m)} \\ &+ (\chi_{V_1} + \chi_{V_3} + \cdots + \chi_{V_{r-2}})\chi_{sym^2\square}^{SO(2N-rm)} + \chi_{\square}^{SO(m)}\chi_{\square}^{SO(2N-rm)}\chi_{V_{\frac{r-1}{2}}},\end{aligned} \tag{53}$$

where $sym^2$ ($asym^2$) means the symmetric (asymmetric) square of representations. When $k + h^\vee = r$, the full character simplifies to

$$PE\left[\frac{(q-q^r)\chi_{adj}^{SO(2N-rm)}}{(1-q)(1-q^r)}\right], \tag{54}$$

which is the same as the vacuum character of the affine $\mathfrak{so}_{2N-rm}$ with level $\tilde{k} = -h^\vee + \frac{h^\vee}{r}$.

**Verification of S duality conjecture**: Consider a theory engineered by the following $(2,0)$ configuration

$$\mathfrak{g} = \mathfrak{sl}_3, \quad \Phi = \frac{T_3}{z^3} + \frac{T_2}{z^3} + \frac{T_1}{z^3}, \quad f = [1^3]. \tag{55}$$

Here $T_i$ are generic diagonal matrices. This theory has one exact marginal deformation, and the flavor symmetry is $U(1)^2 \times SU(3)$. The weakly coupled gauge theory description is found in [64,88]: the original theory is represented by a fourth punctured sphere with three identical $U(1)$ punctures and a $SU(3)$ puncture, while the weakly coupled gauge theory description is found by taking the degeneration limit of punctured sphere as shown in figure 2. The above S duality conjecture suggests that there is a symmetry exchanging three $U(1)$ punctures, and now we will use our index formula to confirm this speculation. The weakly coupled gauge theory is constructed by gauging the $SU(2)$ subgroup of a $D_2[SU(3)]$ theory and a $D_2[SU(5)]$ theory.[9] The 2d VOA of $D_2[SU(3)]$ ($D_2[SU(5)]$) theory is $V^{-3/2}(\mathfrak{sl}_3)$ ($V^{-5/2}(\mathfrak{sl}_5)$). The flavor symmetry of the gauged theory is $SU(3) \times U(1)_a \times U(1)_b$, and the Schur index is

$$\mathcal{I} = \oint \frac{dz}{2\pi i} \left(1 - z^2\right)\left(1 - \frac{1}{z^2}\right) \mathcal{I}^{D_2(SU(3))} \mathcal{I}^V(z;q) \mathcal{I}^{D_2(SU(5))}, \tag{56}$$

with

$$\mathcal{I}^V(z;q) = (q;q)^2 (z^2 q; q)^2 (z^{-2} q; q)^2 \tag{57}$$

being the Schur index of the $SU(2)$ vector multiplet. Expanded in the power series of $q$, the index is

$$\begin{aligned}
\mathcal{I} =&\, 1 + q(2 + \chi^{SU(3)}_{adj}) \\
&+ q^2\left[4 + 2\chi^{SU(3)}_{adj} + \chi^{SU(3)}_{sym^2 adj} + \left(a + b + \frac{1}{ab}\right)\chi^{SU(3)}_{\square} + \left(\frac{1}{a} + \frac{1}{b} + ab\right)\chi^{SU(3)}_{\overline{\square}}\right] + \cdots,
\end{aligned} \tag{58}$$

where two $U(1)$ fugacities are defined as $a = e^{2\pi\alpha(5h_1 + 3h_2)}$ and $b = e^{2\pi\beta(5h_1 - 3h_2)}$ with $h_1$ and $h_2$ being Cartans of two remaining $U(1)$'s in $SU(3)$ and $SU(5)$ after gauging the $SU(2)$. The index is not only symmetric under the exchange of $a$ and $b$, but also symmetric under the permutation of $a$, $b$ and $c$ if we replace $ab$ by $1/c$. One can also see the same symmetry in higher order terms. This fact comes from the permutations symmetry of the three simple punctures in the 6d construction shown in 2. The third $U(1)$ is just formal because the coefficient of the term linear in $q$ indicates that there are only two $U(1)$ currents.

Let us now consider another theory which is engineered by following $(2,0)$ configuration

$$\mathfrak{g} = \mathfrak{sl}_4, \quad \Phi = \frac{T_3}{z^3} + \frac{T_2}{z^3} + \frac{T_1}{z^3}, \quad f = [4]. \tag{59}$$

Here $T_i$ are diagonal matrices. This theory has one exact marginal deformation and the flavor symmetry is $U(1)^3$. This theory is represented by an auxiliary fourth punctured sphere with four identical $U(1)$ punctures. The weakly coupled gauge theory description is found by taking the degeneration limit of this extra punctured sphere, see figure 3 and also [89]. The above S duality picture suggests that there is a symmetry exchanging four $U(1)$ punctures, and we will use the index to confirm this conjecture. The weakly coupled gauge theory description is constructed by gauging the $SU(2)$ subgroup of two $D_2(SU(3))$ theories and a hypermultiplet

---

[9]We summarize different notations of AD theories and their relations together with corresponding references in appendix B.

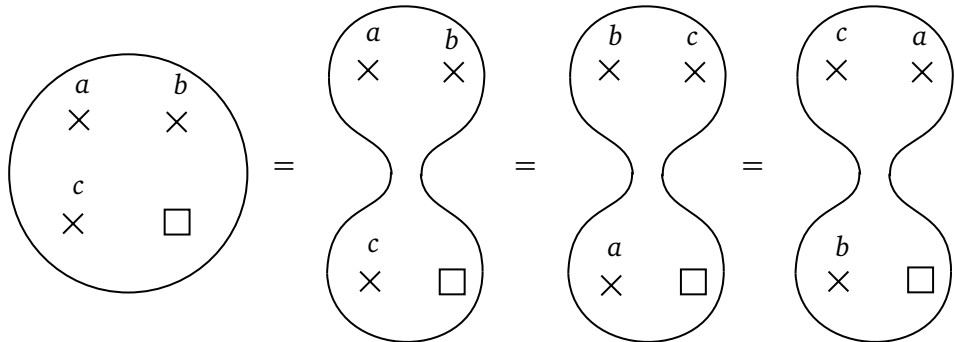

Figure 2: Three Duality frames of theory 55.

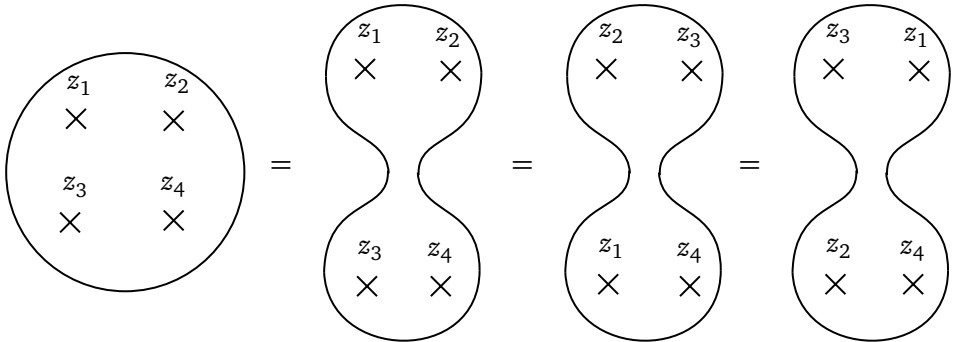

Figure 3: Duality frames of the theory constructed by gluing two $D_2[SU(3)]$ theory and one hypermultiplet.

which transforms as a fundamental of $SU(2)$ gauge group. The flavor symmetry of the gauged theory is $U(1)^3$, and the index is

$$\mathcal{I} = 1 + 3q + \left(\sum_{i=1}^{4}(z_i + z_i^{-1})\right)q^{\frac{3}{2}} + \left(10 + \sum_{1 \le i < j \le 4} z_i z_j\right)q^2 + \cdots, \tag{60}$$

with $z_4 = (z_1 z_2 z_3)^{-1}$. Again, apart from the $S_3$ symmetry among three $U(1)$'s, there is a hidden $S_4$ symmetry coming from the four simple punctures in the 6d construction shown in 3. The fourth $U(1)$ is just formal because the terms linear in $q$ tells us that there are only three conserved currents.

## 4.4 Explicit indices of $(A_1, G)$ theories

Now consider the $(A_1, G)$ theory which can be engineered by the following three-fold singularity

$$f_{ADE}(x, y, z) + w^2 = 0. \tag{61}$$

Here $f_{ADE}$ is a two dimensional $ADE$ singularity. The corresponding VOA for these theories are found in [10, 11, 25, 26, 29] and summarized in table 5. The Schur index can be calculated by

Table 5: The VOA of $(A_1, ADE)$ theories.

| $(G, G')$ | $(J^b[k], f)$ | VOA | $f$ |
|---|---|---|---|
| $(A_1, A_{2N})$ | $(A_1^2[2N+1], f)$ | $\mathcal{W}^{-2+\frac{2}{2N+3}}(A_1, f)$ | $[2]$ |
| $(A_1, A_{2N-1})$ | $(A_{N-1}^N[1], f)$ | $\mathcal{W}^{\frac{-N^2}{N+1}}(A_{N-1}, f)$ | $[N-1, 1]$ |
| $(A_1, D_{2N-1})$ | $(A_1^2[2N-3], f)$ | $\mathcal{W}^{-2+\frac{2}{2N-1}}(A_1, f)$ | $[1, 1]$ |
| $(A_1, D_{2N-2})$ | $(A_{N-1}^N[-1], f)$ | $\mathcal{W}^{\frac{-N^2+2N}{N-1}}(A_{N-1}, f)$ | $[N-2, 1^2]$ |
| $(A_1, E_6)$ | $(A_2^3[4], f)$ | $\mathcal{W}^{-\frac{18}{7}}(A_2, f)$ | $[3]$ |
| $(A_1, E_7)$ | $(A_2^2[3], f)$ | $\mathcal{W}^{-\frac{12}{5}}(A_2, f)$ | $[2, 1]$ |
| $(A_1, E_8)$ | $(A_2^3[5], f)$ | $\mathcal{W}^{-\frac{21}{8}}(A_2, f)$ | $[3]$ |

using our general index formula 39:

$$
(A_1, A_{2N}): \quad \mathcal{I} = PE\left[\frac{q^2 - q^{2N+2}}{(1-q)(1-q^{2N+3})}\right],
$$

$$
(A_1, A_{2N-1}): \quad \mathcal{I} = PE\left[\frac{q + q^2 + (a+a^{-1})q^{\frac{N}{2}} - (a+a^{-1})q^{\frac{N}{2}+2} - q^N - q^{N+1}}{(1-q)(1-q^{N+1})}\right],
$$

$$
(A_1, D_{2N-1}): \quad \mathcal{I} = PE\left[\frac{(q - q^{2N-1})\chi_1(z)}{(1-q)(1-q^{2N-1})}\right],
$$

$$
(A_1, D_{2N-2}):
$$

$$
\mathcal{I} = PE\left[\frac{(1+\chi_1(z))q + (a+a^{-1})\chi_{\frac{1}{2}}(z)q^{\frac{N-1}{2}} - (a+a^{-1})\chi_{\frac{1}{2}}(z)q^{\frac{N+1}{2}} - (1+\chi_1(z))q^{N-1}}{(1-q)(1-q^{N-1})}\right],
$$

$$
(A_1, E_6): \quad \mathcal{I} = PE\left[\frac{q^2 + q^3 - (q^5 + q^6)}{(1-q)(1-q^7)}\right],
$$

$$
(A_1, E_7): \quad \mathcal{I} = PE\left[\frac{q + (a+a^{-1})q^{\frac{3}{2}} + q^2 - q^4 - (a+a^{-1})q^{\frac{9}{2}} - q^5}{(1-q)(1-q^5)}\right],
$$

$$
(A_1, E_8): \quad \mathcal{I} = PE\left[\frac{q^2 + q^3 - (q^6 + q^7)}{(1-q)(1-q^8)}\right].
$$

$$
\tag{62}
$$

Here $a$ is the fugacity for $U(1)$ flavor symmetry and $z$ is the fugacity of $SU(2)$ flavor symmetry, and $\chi_j(z)$ is the character of the spin-$j$ representation of $SU(2)$.

## 4.5 $\tau \to 0$ limit and $a_{4d} - c_{4d}$

The parameter $\tau$ is taking value on the upper half plane. As $\tau$ to zero, the character has the following asymptotic behavior [90]

$$
X_V(\tau) \sim \mathcal{A}(V)e^{\frac{\pi i \mathcal{G}(V)}{12\tau}}, \tag{63}
$$

where $A(V)$ is the amplitude and $G(V)$ is called the asymptotical growth. We have

$$
\mathcal{G}(L_k(\mathfrak{g})) = \left(1 - \frac{h^\vee}{pq}\right)\dim\mathfrak{g}, \quad \mathcal{G}(W^k(\mathfrak{g}, f)) = \mathcal{G}(L_k(\mathfrak{g})) - \dim\mathfrak{f}, \tag{64}
$$

for the level of AKM being $k = -h^\vee + \frac{p}{q}$. The dimension $\dim \mathfrak{f}$ is the dimension of the nilpotent orbit of $f$. On the other hand the above limit of the Schur index is studied in [91], and has the following asymptotic behavior

$$\mathcal{I} \sim e^{-\frac{16\pi^2}{3\beta}(a_{4d} - c_{4d})} \quad \text{as} \quad \beta \to 0. \tag{65}$$

Therefore we have the identification $\beta = \frac{4\pi}{3i}\tau$, and

$$a_{4d} - c_{4d} = -\frac{1}{48}\mathcal{G}, \tag{66}$$

where $\mathcal{G}$ is the growth defined before, see also [31] for the derivation of above formula. Furthermore, the Coulomb branch dictates another relation between $a_{4d}$ and $c_{4d}$ [92]

$$2a_{4d} - c_{4d} = \frac{1}{4}\sum(2u_i - 1), \tag{67}$$

with $u_i$ being the scaling dimension of Coulomb branch operator and the sum runs over all Coulomb branch operators. We can compute $a_{4d} - c_{2d}$ from above Coulomb branch formula, then use it to compare with the answer from index computed from 2d VOA. This would provide a good check for our proposal of the 4d/2d correspondence.

**Example 1**: Consider the case $f = regular$, and take $\mathfrak{g} = ADE$, then the theory can also be engineered by a 3-fold hypersurface singularity

$$f_{ADE}(x, y, z) + w^k = 0. \tag{68}$$

One can find its central charge $a_{4d}$ and $c_{4d}$ using the method presented in [85]. The computation is completely based on the Coulomb branch data, and results are

$$a_{4d} = \frac{l(k-1)(4h^\vee(k+1) + 4k - 1)}{48(h^\vee + k)}, \quad c_{4d} = \frac{l(k-1)(h^\vee(k+1) + k)}{12(h^\vee + k)}, \tag{69}$$

where $l$ is the rank of $\mathfrak{g}$, and $h^\vee$ is the dual Coexeter number. Our VOA is $W^{k'}(\mathfrak{g}, f_{prin})$ algebra with $k' = -h^\vee + \frac{h^\vee}{h^\vee + k}$, therefore (using 64)

$$\mathcal{G} = \frac{l(h^\vee + k) - dim(G)}{h^\vee + k} = \frac{l(k-1)}{h^\vee + k}. \tag{70}$$

One can use the formula in 69 to check that $a_{4d} - c_{4d}$ is indeed $-\frac{1}{48}\mathcal{G}$.

**Example 2**: Consider the case $f = trivial$, so that the 4d SCFT has a $G$ flavor symmetry. The VOA is given by AKM $V^{k'}(\mathfrak{g})$ with $k' = -h^\vee + \frac{h^\vee}{h^\vee + k}$. The 4d central charges $a_{4d}$ and $c_{4d}$ are

$$a_{4d} = \frac{(4h^\vee + 4k - 1)(h^\vee + k - 1)}{48(h^\vee + k)}\dim\mathfrak{g}, \quad c_{4d} = \frac{1}{12}(h^\vee + k - 1)\dim\mathfrak{g}. \tag{71}$$

The growth (use 64) is

$$\mathcal{G} = \frac{h^\vee + k - 1}{h^\vee + k}\dim\mathfrak{g}, \tag{72}$$

which is again the same as $-48(a_{4d} - c_{4d})$.

# 5 Zhu's $C_2$ algebra and the ring of the Schur sector

Given a 2d VOA $V$, one can associate an associative and commutative ring $R_V$ which is called Zhu's $C_2$ algebra. $R_V$ is in general an affine scheme and one can get a reduced affine ring which is further identified with the Higgs branch chiral ring of the corresponding 4d theory. It is quite interesting to consider the reduced affine ring as one can learn the structure of Higgs branch chiral ring [29, 31, 67].

The new perspective of this paper states that the Zhu's $C_2$ algebra is actually more important than its reduced counterpart. On the one hand, if a 4d SCFT has no Higgs branch, the reduced affine ring is just trivial, however, the Zhu's $C_2$ algebra can still be quite non-trivial in this case. On the other hand, there are many 4d SCFT which shares the same Higgs branch so reduced affine rings of their associated VOAs are the same, however, their Zhu's $C_2$ algebras can still be very different, therefore the Zhu's $C_2$ algebra can be used to distinguish different 4d $\mathcal{N} = 2$ SCFTs. Moreover, the Zhu's $C_2$ algebra can be thought as the classical limit of Zhu's algebra whose representation theory is closely related to the representation theory of the VOA, so the non-reduced version is definitely more important than its reduced version.

Since the reduced affine ring of Zhu's $C_2$ algebra gives the Higgs branch chiral ring, we might call Zhu's algebra as the ring of the Schur sector. The main purpose of this section is to compute explicitly Zhu's $C_2$ algebras for VOAs considered in this paper. The physical meaning of this ring seems quite interesting. For example the reduced ring from the Schur ring, which gives the coordinate ring of the Higgs branch, is related to the free field description [53]. More details on this relation will be explained in the subsequent paper [93].

## 5.1 Zhu's $C_2$ algebra and Jacobi algebra

Consider $W^{k'}(\mathfrak{g}, f)$ algebra with $f$ being principal, so there is no flavor symmetry left. Recall $k' = -h^\vee + \frac{h^\vee}{h^\vee + k}$. The corresponding $W$ algebra is just the minimal model of $W$-algebra. The character of the vacuum module takes the following form

$$\mathrm{ch}_{W^{k'}(g, f_{prin})}(\tau) = PE\left[\frac{\sum_i q^{d_i} - q^{h^\vee + k + 1}(\sum q^{-d_i})}{(1 - q)(1 - q^{h^\vee + k})}\right], \quad k' = -h^\vee + \frac{h^\vee}{h^\vee + k}. \tag{73}$$

This $W$-algebra is strongly generated by a set of fields $W_i$ with scaling dimension $d_i$ which are just degrees of Casimirs of the Lie algebra $\mathfrak{g}$. These fields also generate the Zhu's algebra as the VOA is strongly finitely generated. From the character, one finds that there are $l$ singular vectors with scaling dimension $h^\vee + k + 1 - d_i$, $i = 1, \ldots, l$. To write down explicitly its Zhu's $C_2$ algebra, one usually needs to analyze its singular vectors explicitly.

However we will take a different approach for $f$ being principal and discover a surprising appearance of the singularity theory and the Jacobi algebra. Since there is no flavor symmetry left, and it is believed that the Higgs branch is trivial, the Zhu's $C_2$ algebra is finite dimensional. Based on some concrete examples, we would like to conjecture that the Zhu's $C_2$ algebra of a minimal $W$-algebra is isomorphic to the Jacobi algebra of a quasi-homogeneous isolated singularity $F$.

Let us first review the associated Jacobi algebra of a quasi-homogeneous isolated singularity $F$. Consider a hypersurface singularity defined by a polynomial $F : (\mathbb{C}^n, 0) \to (\mathbb{C}, 0)$ with a $\mathbb{C}^*$ action

$$F(\lambda^{\omega_i} z_i) = \lambda^d F(z), \tag{74}$$

then one has the weight data associated with this singularity $F$ as $(w_1, \ldots, w_n; d)$, and one might choose a particular normalization such that these weights are all integers (we do not require that they are pairwise co-prime though). The singularity is isolated if equations $F = \frac{\partial F}{\partial z_i} =$

0 have a unique solution at $z_i = 0$. The Jacobi algebra $J_F$ associated with $F$ is then defined as

$$J_F = \mathbb{C}[z_1, \ldots, z_n] \bigg/ \left\{ \frac{\partial F}{\partial z_1}, \ldots, \frac{\partial F}{\partial z_n} \right\}. \tag{75}$$

On the other hand, given a set of integral weights $(w_1, \ldots, w_n; d)$, one might try to construct a polynomial $F$ such that $F$ has an isolated singularity at the origin. A necessary condition for this to happen is that for each variable $z_i$, there is at least one monomial of the following form

$$\{z_1 z_i^a, \ldots, z_n z_i^a\}. \tag{76}$$

The degree $d$ and weights $w_i$ provide a constraint on whether such monomial is possible or not.

Coming back to our problem of finding Zhu's $C_2$ algebras of $W$-algebra minimal models, we have already learned that the Zhu's algebra is generated by elements $W_i, i = 1, \ldots, l$ with scaling dimension $d_i$. From the index, we have singular vectors with scaling dimension $h^\vee + k + 1 - d_i$, $i = 1, \ldots, l$, and we conjecture that these singular vectors are enough to generate all relations, so the Zhu's algebra might take the following form

$$\mathbb{C}[W_1, \ldots, W_l]/\{F_1, \ldots, F_l\}. \tag{77}$$

Each $W_i$ has degree $d_i$ and $F_i$ has degree $h^\vee + k + 1 - d_i$. We would like now conjecture that the Zhu's algebra is isomorphic to a Jacobi algebra associated with a hypersurface singularity $F$ of type

$$(d_1, \ldots, d_l; h^\vee + k + 1). \tag{78}$$

To find the explicit form of $F$, we simply write down the possible monomials from the set 76 for each variable $W_i$.

**Example 1**: Take $\mathfrak{g} = \mathfrak{sl}_3$ and $k = 4$. The Zhu's algebra is generated by two variables $W_2, W_3$ with weights $(2, 3; 8)$. The monomials in $F$ should be degree 8, then for $W_2$ variable, we have a monomial $W_2^4$ which is in the set 76, and for variable $W_3$, we have a monomial $W_2 W_3^2$. So the polynomial $F = W_2^4 + W_2 W_3^2$, the Jacobi algebra is then

$$\frac{\mathbb{C}[W_2, W_3]}{\{W_2^3 + W_3^2, W_2 W_3\}}. \tag{79}$$

**Example 2**: For $\mathfrak{g} = \mathfrak{sl}_n$ and arbitrary $k$, we have the level-rank duality so that the same theory can be realized by $\mathfrak{g} = \mathfrak{sl}_k$ with the other data $n$. The associated Zhu's algebra of two descriptions should be equivalent. In this example, we show that this is indeed the case. Take $n = 3, k = 4$, and we have computed Zhu's algebra using the $\mathfrak{g} = \mathfrak{sl}_3$ description in the previous example. Using the $\mathfrak{sl}_4$ description, we have three generators $W_2, W_3, W_4$ with weights $(2, 3, 4; 8)$, and the degree $d$ of monomials in $F$ should be 8. The polynomial is then $F' = W_2^4 + W_2 W_3^2 + W_4^2$, and it is well known that the Jacobi algebra of $F'$ is the same as the Jacobi algebra associated with the polynomial $F = W_2^4 + W_2 W_3^2$.

## 5.2 Singular vector and general proposal for Zhu's $C_2$ algebra

Once we know the singular vector of a VOA, we can actually compute the Zhu's $C_2$ algebra from the definition in section 2.1. Here we give some explicit examples.

**Example 1**: Consider the $(A_1, A_{2N})$ AD theory whose VOA is the $(2, 2N+3)$ minimal model of the Virosora algebra. This VOA is strongly generated by the energy-momentum tensor $T(z)$. The first non-trivial singular vector appears at scaling dimension $2N + 2$ with the following form

$$[(L_{-2})^{N+1} + \ldots]|0\rangle, \tag{80}$$

where we ignore terms involving operators $L_n$ with $n < 3$ as these terms give derivative fields, which are in the same class as $[0]$ in the Zhu's $C_2$ algebra. In the VOA literature, the scaling dimension does not enter the expansion of fields, i.e. $a(z) = \sum_n a_n z^{-n-1}$, and one also use this convention for the stress tensor field $T(z)$. However, in physics literature, one usually use the convention $T(z) = \sum_n L_n z^{-n-2}$, so we have the identification $a_n = L_{n-1}$. Now in physics convention, the Verma module is generated by following vector

$$L_{j_1} \dots L_{j_m} |0\rangle, \quad j_1 \le \dots j_m \le 2. \tag{81}$$

The operator-states correspondence is

$$L_{j_1} \dots L_{j_m} |0\rangle, \to a_{j_1 \dots j_m}(z) =: \partial_z^{-j_1-2} T(z) \dots \partial_z^{-j_m-2} T(z) : . \tag{82}$$

In particular $L_{-2}|0\rangle \to T(z)$, and $L_{-n}|0\rangle \to \partial^{n-2}T(z)$. If we expand the field $T(z) = \sum_n a_n z^{-n-1}$, then $(\partial^{n-2}T(z))_{-2} = a_{-n} = L_{-n-1}$. This implies that $L_{-n}$ with $n > 3$ is the $-2$ modes of a vector in VOA. Now the $C_1$ subspace of our VOA (using the $-1$ shift in the mode expansion) is as

$$C_1(V) = \{a_{-2}b | a, b \in V\}. \tag{83}$$

So a state vector in 81 involves the raising operator $L_{-n}$ with $n < -3$ would be in $C_1(V)$ and is zero in the Zhu's $C_2$ algebra $V/C_1(V)$. The generator of Zhu's $C_2$ algebra is the state $L_{-2}|0\rangle \to T(z)$, and the singular vector 80 gives a relation

$$T^{N+1} = 0, \tag{84}$$

where 0 means the fields involve derivatives. So Zhu's $C_2$ algebra of $(2, 2N+3)$ minimal model is simply

$$\mathbb{C}[T]/T^{N+1}. \tag{85}$$

**Example 2:** Let us now consider theory with $\mathfrak{g} = \mathfrak{sl}_{l+1}$, and the level $k' = -\frac{l+1}{2}$, and $f$ is trivial. We take $l$ to be even, and this theory is also called $D_2 SU(l+1)$. This theory has Coulomb branch spectrum $[\frac{3}{2}, \frac{5}{2}, \dots, \frac{l+1}{2}]$, and flavor symmetry is $SU(l+1)$. The singular vector which generate the maximal proper ideal is found in [94]

$$v = \left(\sum_{i=1}^{l} \frac{l-2i+1}{l+1} h_i(-1)e_\theta(-1) - \sum_{i=1}^{l} e_{\epsilon_1-\epsilon_{i+1}}(-1)e_{\epsilon_{i+1}-\epsilon_{l+1}}(-1) - \frac{1}{2}(l-1)e_\theta(-2)\right)|0\rangle. \tag{86}$$

Here $\theta$ is the longest root of Lie algebra $\mathfrak{sl}_{l+1}$ and $\theta = \alpha_1 + \dots + \alpha_l$ with $\alpha_i$ the set of simple roots. We use the convention that $\alpha_i = \epsilon_i - \epsilon_{i+1}$ are simple roots. So $\epsilon_1 - \epsilon_{i+1} = \alpha_1 + \dots + \alpha_i$ and $\epsilon_{i+1} - \epsilon_{l+1} = \alpha_{i+1} + \dots + \alpha_l$. The level of this singular vector is 2, and it is the highest weight of the adjoint representation which matches the $-q^2\chi(z)$ term in the index formula 34 ($h^\vee + k$ is 2 in this case).

**Example 3:** Consider $D_2[SU(3)]$ theory, and the corresponding VOA is $V^{-\frac{3}{2}}(\mathfrak{sl}_3)$. The VOA is strongly generated by the fields $x_i \in \mathfrak{sl}_3$, so the corresponding Zhu's $C_2$ algebra is also generated by $x_i$. We choose the basis of the Lie algebra to be $(h_1, h_2, e_{12}, e_{23}, e_{13}, e_{21}, e_{32}, e_{31})$, where $h_1$ and $h_2$ generate the Cartan subalgebra, and $e_{ij}$ is the root vector of the root $\epsilon_i - \epsilon_j$. From our general index formula, we can see the image of the maximal proper ideal of $R_{V^{-\frac{3}{2}}(\mathfrak{sl}_2)}$

is an adjoint $\mathfrak{sl}_3$-module. We write down this ideal $I$ explicitly

$$\bar{v} = \frac{1}{3}(h_1 - h_2)e_{13} + e_{12}e_{23},$$

$$(\mathrm{ad}e_{21})\bar{v} = -\frac{1}{3}(2h_1 + h_2)e_{23} + e_{21}e_{13},$$

$$(\mathrm{ad}e_{32})\bar{v} = -\frac{1}{3}(h_1 + 2h_2)e_{12} - e_{13}e_{32},$$

$$(\mathrm{ad}e_{32})(\mathrm{ad}e_{21})\bar{v} = -e_{12}e_{21} + e_{13}e_{31} + \frac{1}{3}(2h_1 + h_2)h_2,$$

$$(\mathrm{ad}e_{21})(\mathrm{ad}e_{32})\bar{v} = -e_{23}e_{32} + e_{13}e_{31} + \frac{1}{3}(h_1 + 2h_2)h_1, \tag{87}$$

$$(\mathrm{ad}e_{21})(\mathrm{ad}e_{32})(\mathrm{ad}e_{21})\bar{v} = \frac{1}{3}(h_1 + 2h_2)e_{21} + e_{23}e_{31},$$

$$(\mathrm{ad}e_{32})(\mathrm{ad}e_{21})(\mathrm{ad}e_{32})\bar{v} = \frac{1}{3}(2h_1 + h_2)e_{32} - e_{31}e_{12},$$

$$(\mathrm{ad}e_{21})(\mathrm{ad}e_{32})(\mathrm{ad}e_{21})(\mathrm{ad}e_{32})\bar{v} = \frac{1}{3}(h_1 - h_2)e_{31} + e_{32}e_{21}.$$

Here $h_i$, $i = 1, 2$ and $e_{ij}, i \neq j$ generate the polynomial ring $\mathbb{C}[h_i, e_{ij}]$ and then

$$R_{V^{-\frac{3}{2}}(\mathfrak{sl}_2)} = \mathbb{C}[h_i, e_{ij}]/I. \tag{88}$$

To get $X_{V^{-\frac{3}{2}}(\mathfrak{sl}_2)}$, one first work out the radical $I^r$ of $I$, which is

$$\begin{aligned}
&(h_1 - h_2)e_{13} + 3e_{12}e_{23}, \\
&(2h_1 + h_2)e_{23} - 3e_{21}e_{13}, \\
&(h_1 + 2h_2)e_{12} + 3e_{13}e_{32}, \\
&h_1^2 + 4e_{12}e_{21} + e_{13}e_{31} + e_{23}e_{32}, \\
&h_1 h_2 - 2e_{12}e_{21} + e_{13}e_{31} - 2e_{23}e_{32}, \\
&h_2^2 + e_{12}e_{21} + e_{13}e_{31} + 4e_{23}e_{32}, \\
&(h_1 + 2h_2)e_{21} + 3e_{23}e_{31}, \\
&(2h_1 + h_2)e_{32} - 3e_{31}e_{12}, \\
&(h_1 - h_2)e_{31} + 3e_{32}e_{21},
\end{aligned} \tag{89}$$

and $X_{V^{-\frac{3}{2}}(\mathfrak{sl}_2)} = \mathrm{spec}\left[\mathbb{C}[h_i, e_{ij}]/I^r\right]$. One can show that $X_{V^{-\frac{3}{2}}(\mathfrak{sl}_2)}$ is isomorphic to $\overline{\mathbb{O}}_{\min}$ and has dimension 4. Moreover, the ideal $I^r$ has the same representation structure as the Joseph ideal of $\mathfrak{sl}_3$.

## 5.3 A general proposal for Zhu's $C_2$ algebra

Now, we would like to make a general conjecture of Zhu's $C_2$ algebras of our $W$-algebras $W^{k'}(\mathfrak{g}, f)$. The relation between Zhu's $C_2$ algebras $f = trivial$ and that of more general $f$ through qDS reduction was also discussed in [68]. Recall that given a nilpotent element $f$ and its associated $\mathfrak{sl}_2$ triple, we have a subgroup $SU(2) \times G_F$ with $G_F$ the flavor symmetry group. The adjoint representation of $\mathfrak{g}$ is decomposed into representations of $SU(2) \times G_F$ as $adj_{\mathfrak{g}} \to \oplus_j (V_j \otimes R_j)$. Given the structure of our simplified index formula, we would like to conjecture that the Zhu's $C_2$ algebra of our VOA has the following form:

1. Zhu's algebra is generated by the fields $W_j$ with scaling dimension $(1+j)$, and the number of such fields are given by $\dim R_j$, and they transform as $R_j$ representation of flavor group $G_j$.

2. Relations for fields $W_j$'s are generated only by the set of singular vectors $v_j$ at scaling dimension $q^{k+h^\vee-j}$ in the representation $R_j$.

3. The associated variety is simply $S_f \cap O_{k'}$ [78]. Here $S_f$ is the Slodowy slice of $f$ and $O_{k'}$ is the nilpotent orbit which is determined by the level $k'$.

The above proposal is based on the structure of character formula presented in 39. The detailed form of the ideal is quite complicated, and we do not know a systematical way of writing down the ideal.

Here we give some conjectured form of Zhu's $C_2$ algebra for $(A_1, G)$ theories. Zhu's $C_2$ algebras for theories with no flavor symmetries can be decided using the method proposed in section 5.1.

$$
\begin{aligned}
(A_1, A_{2N}): \quad & R_V = J_I, \quad I = T^{N+2}, \\
(A_1, E_6): \quad & R_V = J_I, \quad I = T^4 + TW^2, \\
(A_1, E_8): \quad & R_V = J_I, \quad I = W^3 + TW^3.
\end{aligned}
\tag{90}
$$

Here $R_V$ is the Zhu's $C_2$ algebra and $J_I$ is the associated Jacobi algebra of polynomial $I$. $T$ has scaling dimension two, and $W$ has scaling dimension three.

Zhu's $C_2$ algebras for theories with one $U(1)$ flavor symmetry are computed as follows (for $(A_1, A_{2N-1})$ theories)

$$
(A_1, A_{2N-1}): \quad \mathbb{C}[J, T, W_+, W_-]/I,
$$

$$
\text{with } I = \left\{ W_+(J^2+T), W_-(J^2+T), W_+W_- + \sum_{i=0}^{N-2i\geq 0} J^{N-2i}T^i, W_+W_-J + \sum_{i=0}^{N-2i\geq 0} J^{N+1-2i}T^i \right\},
\tag{91}
$$

where $J$ has scaling dimension one. $T$ has scaling dimension two, and $W_\pm$ have scaling dimension $\frac{N}{2}$ and $U(1)$ charge $\pm 1$. And for $(A_1, E_7)$ theory, we have:

$$
\begin{aligned}
(A_1, E_7): \quad & R_V = \mathbb{C}[J, W_+, W_-, T]/I, \\
& \text{with} \\
& I = \{J^4 + J^2T + T^2 + JW_+W_-, W_+(J^3 + JT + W_+W_-), \\
& \qquad W_-(J^3 + JT + W_+W_-), J^5 + J^3T + JT^2 + W_+W_-(J^2+T)\},
\end{aligned}
\tag{92}
$$

here $J$ has scaling dimension one, and $W$ has scaling dimension two, and $W_\pm$ has scaling dimension $\frac{3}{2}$.

And the Zhu's $C_2$ algebras for theories with at least one $SU(2)$ ($U(2)$) flavor symmetry are computed as follows

$$
(A_1, D_{2N-2}): \quad R_V = \mathbb{C}[J_j^i, W_i^+, W_i^-]/I,
$$

$$
\text{with } I = \left\{ (\delta_i^j \operatorname{tr} J + J_i^j)W_j^+, (\delta_i^j \operatorname{tr} J + J_i^j)W_j^-, W_i^+W_j^- + \left( \sum_{l=0}^{N} (\operatorname{tr} J^l)(J^{N-l})_i^k \right) \epsilon_{kj} \right\}.
\tag{93}
$$

Here dimension one operators $J_j^i$'s are in the adjoint representation of $U(2)$ with $i, j = 1, 2$. Dimension $\frac{n-1}{2}$ operators $W_i^\pm$ form fundamental and antifundamental representations of $U(2)$ respectively. In those cases, the ideal is found by following strategy: a): since we know the grading of each polynomial $f_i$ in the ideal, we first write all possible combinations of monomials in $f_i$; b): we also know the Higgs branch dimension in each case, i.e. $n_h = 1$ for $(A_1, A_{2N-1})$ and $(A_1, E_7)$ case, and $n_h = 2$ for $(A_1, D_{2N-2})$ case. So a consistent condition is that the dimension of the reduced ring should be equal to the dimension of the Higgs branch.

# 6  Kazhdan filtration and Macdonald index

The Schur index of a $4d$ $\mathcal{N} = 2$ SCFT has only one fugacity $q$, and it is identified with the character of the vacuum module of its associated 2d VOA. The Macdonald index also counts Schur operators, but with two fugacities $q$ and $T$, see section 2. Since the 4d/2d correspondence is actually between the Schur sector and the VOA itself, it should be possible to recover the Macdonald index from the structure of VOA. The VOA has one natural grading which is just the eigenvalue of zero mode of 2d energy-momentum tensor $T(z)$. To recover the Macdonald index, one need to find another grading. Such grading is found for some AD theories in [22]. Here we generalize their results to all $W$-algebras $W^{k'}(\mathfrak{g}, f)$ considered in this paper, and a crucial ingredient is a new type of filtration called Kazhdan filtration defined for our W algebra.

First let us consider the universal affine VOA $V^k(\mathfrak{g})$ associated with an AKM algebra. For a Lie algebra $\mathfrak{g}$, one has its universal enveloping algebra $\mathcal{U}(\mathfrak{g})$ [95]. By the PBW theorem, $V^k(\mathfrak{g})$ has a PBW basis consisting of monomials of the form:

$$x_{n_1}^{i_1} \ldots x_{n_m}^{i_m} |0\rangle, \tag{94}$$

where $n_1 \le n_2 \ldots \le n_m < 0$, and if $n_j = n_{j+1}$, then $i_j \le i_{j+1}$. Here $x^i$ is an ordered basis of Lie algebra $\mathfrak{g}$. The universal affine VOA $V^k(\mathfrak{g})$ is then isomorphic to its universal enveloping algebra $\mathcal{U}(\mathfrak{g})$. The PBW filtration on the VOA is then defined as follows

$$K_{-1}\mathcal{U}(g) = 0, \quad K_0\mathcal{U}(g) = \mathbb{C}, \quad K_p\mathcal{U}(g) = \mathfrak{g}F_{p-1}\mathcal{U}(g) + F_{p-1}\mathcal{U}(g). \tag{95}$$

Here $F_p\mathcal{U}(g)$ is the PBW filtration on $\mathcal{U}(g)$. The VOA is generated by $\dim(\mathfrak{g})$ fields $x_i(z)$ and all states are spanned by derivatives of these generating fields. The PBW filtration at level $p$ simply includes those states with at most $p$ generating fields $x^i(z)$ (the number of derivatives on those fields is not limited though), i.e., one assign T grading one to the fundamental fields $x^i(z)$. Since each $x_i(z)$ gives a $\hat{\mathcal{B}}_1$ type operators, its $T$ grading is just one, and it is natural to identify PBW grading as the one giving $T$ grading. This is almost right, but there is an important subtly that we will discuss later.

Now consider $W$-algebra $W^k(\mathfrak{g}, f)$. Given an $\mathfrak{sl}_2$ triple $(x, e, f)$, the Lie algebra has the following decomposition

$$\mathfrak{g} = \oplus_{j \in \mathbb{Z}/2} \mathfrak{g}_j. \tag{96}$$

For each element $v_j$ in $\mathfrak{g}_j^f$, one has a generator $J_{v_j}$ in VOA with scaling dimension $1 + j$ (for $j \ge 0$). Now set $F_p\mathcal{U}(\mathfrak{g})[j] = \{[x, u] = ju, u \in \mathcal{U}_p(\mathfrak{g})\}$, one then has the following (grading by half-integer) Kazhdan filtration

$$K_p\mathcal{U}(\mathfrak{g}) = \sum_{i+j \le p} F_i\mathcal{U}(\mathfrak{g})[j]. \tag{97}$$

In particular, the generator $J_{v_j} \in \mathfrak{g}_j^f$ is in $F_1\mathcal{U}(\mathfrak{g})$ and $j$ value is just $j$, so it is in space $K_s\mathcal{U}(\mathfrak{g})$ with $s \ge j + 1$.

We need to modify a little bit on above Kazhdan filtration so that this grading can give us Macdonald index. Since our VOA is strongly generated by the fields $J_{v_j}$, we can assign a grading to VOA by using the grading of the generators from Kazhdan filtration. The rule is following: since $J_{v_0}$ gives $\hat{\mathcal{B}}_1$ type operator, we assign $T$ grading one to it. For the other fields $J_{v_j}$ though, we have to assign grading $j$ to it (from Kazhdan grading). Using this modified grading, we now have an increasing filtration on our VOA (which we still call Kazhdan filtration):

$$K_0 \subset K_1 \subset K_2 \subset \ldots \tag{98}$$

and a decreasing Li's filtration (see section 2)

$$F_0 \supset F_1 \supset F_2 \supset \ldots \tag{99}$$

and we can form the following double graded space

$$H^{p,k} = \frac{F^p \cap K_k}{F^{p+1} \cap K_k} + K_{k-1}. \tag{100}$$

And each vector in subspace $H^{p,k}$ has two gradings: one grading is just the Kazhdan grading $k$, and other one is the conformal grading $\Delta$ (notice that $\Delta$ is different from $p$). Using above double grading, we define Macdonald index as follows:

$$\mathcal{I}(q,T) = \sum_{H^{p,k}} q^{\Delta - \frac{c}{24}} T^k. \tag{101}$$

In the above consideration, the Kazhdan filtration is defined by choosing a generating set. In particular, in the AKM case, we choose $x^i(z)$ (with $x^i$ generating Lie algebra $\mathfrak{g}$) as the generating set. Now according to Sugawara construction, the energy momentum tensor $T(z) \sim x(z)^2$. If we use the Kazhdan filtration with $x^i(z)$ as generating set, $x^2$ would be in $K_2$ and has $T$ grading two which is inconsistent with the fact that the $T$ grading of $T(z)$ field should be just one. To remedy this situation, we use a strategy following [22]: we add $T(z)$ to our generating set and assign $T$ grading one to it, and imposing the relation $T(z) \sim x(z)^2$. Now VOA is strongly generated by the fields $x^i(z)$ and $T(z)$, and we have a similar Kazhdan filtration using the grading of generating fields. Now $T(z)$ contributes $q^2 T$ to the index, however, the relation $T(z) - x(z)^2 = 0$ is actually in space $K_2$ and actually contributes $-q^2 T^2$ to the index.

**Example**: Consider $(A_1, D_{2N-1})$ theory. The corresponding VOA is $V_{-2+\frac{2}{2N-1}}(\mathfrak{sl}(2))$, and it has a singular vector at level $2N-1$ transforming in adjoint representation, which actually contributes to index a term $-q^{2N-1} T^N \chi_{adj}(z)$. The Macdonald index then has the following form

$$\mathcal{I}(q,T) = PE\Big[\frac{qT\chi_{adj}(z) + q^2 T - q^2 T^2 - q^{2N-1} T^N \chi_{adj}(z)}{(1-q)(1-q^{2N-1})} + f(q,T,z)\Big]. \tag{102}$$

Here $f(q,1,z) = 0$ so that we recover Schur index in $T = 1$ limit. In the above computation, Zhu's $C_2$ algebra actually plays a crucial role. Let us take $N = 2$ for an example. We use a generating set $(a,b,c,T)$ for our VOA so that the Zhu's $C_2$ algebra is also generated by these fields. The Zhu's $C_2$ algebra is

$$\frac{C[a,b,c,T]}{\{T - (ab + c^2), aT, bT, cT\}}, \tag{103}$$

where the ideal has four generators. The first one contributes to the index with a term $-q^2 T^2$, and the last three contribute with a term $-q^3 T^2 \chi_{adj}(z)$.

We do not have the closed formula for the Macdonald index for the general case, but we do know first few terms (in the case of AKM, one need to add the contribution from energy momentum tensor.)

$$\mathcal{I}(q,T) = PE\left[\frac{q\sum q^j T^{j+[\frac{h^\vee - j}{h^\vee}]}\chi_{R_j}(z) - (qT)^{h^\vee + k}\sum q^{-j} T^{-[\frac{h^\vee + k - j}{h^\vee}] - j}\chi_{R_j}(z)}{(1-q)} + \ldots\right]. \tag{104}$$

Here $R_j$ is the representation of flavor symmetry group $G_F$, and $[a]$ means that we take the integral part of the number $a$ inside the square bracket.

# 7 Conclusion

It is quite difficult to understand the Schur sector of a general $\mathcal{N} = 2$ SCFT due to the fact that most of these theories are strongly coupled, and no powerful tools such as the Seiberg-Witten geometry of the Coulomb branch is available. However, the correspondence between 4d $\mathcal{N} = 2$ SCFTs and 2d VOAs makes understanding of the Schur sector possible when the corresponding 2d VOA is known. In the previous work [57], we have identified the associated 2d VOAs for a large class of 4d $\mathcal{N} = 2$ SCFTs constructed from 6d $(2, 0)$ theories. In this paper, we use the knowledge of 2d VOA to learn lots of interesting properties of the Schur sector:

- The Schur index is computed from the vacuum character of $W$-algebra and can be put in a surprisingly simple form.

- The associated Zhu's $C_2$ algebra can be computed from the VOA and can be regarded as the ring associated with the Schur sector.

- One can use the Kazhdan filtration of the $W$-algebra to compute the Macdonald index.

The Zhu's $C_2$ algebra can be thought as the associated ring of the Schur sector. This algebra is in general quite complicated, and for some subset of theories, we found a surprising isomorphism between the Zhu's $C_2$ algebra and the Jacobi algebra of a hypersurface singularity. It would be interesting to further check our proposal. It would be also interesting to further study the physical meaning of this algebra. Moreover, one can have an associative (but not commutative) algebra which is called Zhu's algebra. This algebra controls lots of interesting information of the representation theory of the VOA, and it would be interesting to understand the physical meaning of Zhu's algebra as well.

We now would like to make some speculations on generators of Schur sector ring of models considered in this paper. The generator has the following contribution to Macdonald index $q^{1+j} T^j$, $j > 0$ and $qT$ for $j = 0$. Given a Schur operator $\hat{\mathcal{C}}_{R,(j_1,j_2)}$, its contribution to Macdonald index is $q^{2+R+j_1+j_2} T^{1+R+j_2-j_1}$ (see 3). Since there are three independent quantum numbers for a Schur operator and the Macdonald index can only capture two quantum numbers, we can not completely determine the Schur operator type corresponding to the generators. However, based on some known examples. we make the following conjecture: all generators are just scalars $\hat{\mathcal{B}}_{R+1} = \hat{\mathcal{C}}_{R,(-1/2,-1/2)}$ and $\hat{\mathcal{C}}_{R,(0,0)}$. It would be interesting to verify this conjecture.

One can construct new $\mathcal{N} = 2$ SCFTs by gauging AD theories (They are called AD matters) considered in this paper, and the corresponding VOA of the gauged theory can be found from the cosets of $W$-algebras associated with AD matter [57]. The Schur index of the gauged system can also be computed using the index of the AD matters. We have used this strategy to check S duality conjecture proposed in [39, 58, 59]. Now one could have a different S duality where new AD matter would appear, and these AD matter is not included in theories considered in this paper. However, using the index of the full theory found from the original S duality frame and known results of other components in this new duality frame, it is possible to find indices of these new AD matters. Using the above method, it might be possible to get Schur indices of all AD theories constructed from 6d $(2, 0)$ theories.

In this paper, we mainly focus on the vacuum module of $W$-algebra. Other modules also play important roles in the study of 4d $\mathcal{N} = 2$ SCFTs [24, 25, 37]. These modules will be studied in a forthcoming paper [96].

## Acknowledgements

Authors would like to thank Peng Shan for helpful discussions. DX and WY are supported by Yau mathematical sciences center at Tsinghua University. WY is also supported by the young overseas high-level talents introduction plan.

## A  Notation of special functions

The convention of special functions are summarized in this section. First the definition of the q-Pochhammer symbol $(x;q)$ is

$$(x;q) = \prod_{i=0}^{\infty}(1-xq^i), \tag{105}$$

and $(x;q)_i$ is defined as

$$(x;q)_i = \frac{(x;q)}{(xq^i;q)}. \tag{106}$$

The definition of the plethystic exponential is

$$PE[f(a,\cdots)] = e^{\sum_{n=1}^{\infty}\frac{1}{n}f(a^n,\cdots)}, \tag{107}$$

therefore

$$PE[x] = e^{\sum_{n=1}^{\infty}\frac{1}{n}x^n} = \frac{1}{1-x}, \tag{108}$$

and

$$PE\left[\frac{x}{1-q}\right] = e^{\sum_{n=1}^{\infty}\frac{1}{n}\frac{x^n}{1-q^n}} = \prod_{i=0}^{\infty}\frac{1}{1-xq^i} = \frac{1}{(x;q)}. \tag{109}$$

The $\eta$-function $\eta(\tau)$ is defined as

$$\eta(\tau) = q^{\frac{1}{24}}\prod_{n=1}^{\infty}(1-q^n) = q^{\frac{1}{24}}(q;q), \tag{110}$$

with $q = e^{2\pi i\tau}$. And four Jacobian theta functions are defined as

$$\begin{aligned}
\theta_{00}(z,\tau) &= q^{-\frac{1}{24}}\eta(\tau)\prod_{n=1}^{\infty}(1+e^{2\pi iz}q^{n-\frac{1}{2}})(1+e^{-2\pi iz}q^{n-\frac{1}{2}}), \\
\theta_{01}(z,\tau) &= q^{-\frac{1}{24}}\eta(\tau)\prod_{n=1}^{\infty}(1-e^{2\pi iz}q^{n-\frac{1}{2}})(1-e^{-2\pi iz}q^{n-\frac{1}{2}}), \\
\theta_{10}(z,\tau) &= q^{\frac{1}{12}}e^{-\pi iz}\eta(\tau)\prod_{n=1}^{\infty}(1+e^{-2\pi iz}q^n)(1+e^{2\pi iz}q^{n-1}), \\
\theta_{11}(z,\tau) &= -iq^{\frac{1}{12}}e^{-\pi iz}\eta(\tau)\prod_{n=1}^{\infty}(1-e^{-2\pi iz}q^n)(1-e^{2\pi iz}q^{n-1}).
\end{aligned} \tag{111}$$

One can also rewrite all theta functions in terms of q-Pochhammers or plethystic exponentials, which plays an important role in the main text.

# B Hitchin system descriptions for $(G, G')$ and $D_p(G)$ theory

There are various class of 4d $\mathcal{N} = 2$ AD SCFTs found in the literature, and they have different labels which might cause some confusions. Here we provide a mapping between these labels and our theories. There are three class of theories:

1. Theories with label $(G, G')$ [97]. This class of theories are engineered by following 3-fold singularity:
$$f_G(x, y) + f_{G'}(z, w) = 0. \tag{112}$$

   Here $G = ADE$ and $f_G(x, y)$ are following polynomials:

$$f_{A_N} = x^2 + y^{N+1}, \; f_{D_N} = x^{N-1} + xy^2, \; f_{E_6} = x^3 + y^4, \; f_{E_7} = x^3 + xy^3, \; f_{E_8} = x^3 + y^5. \tag{113}$$

   There is a symmetry exchanging $G$ and $G'$ in the definition of the 3d singularity so that the $(G, G')$ theory is the same as the $(G', G)$ theory. This class of theories include the original AD theory found in [98] (It is the $(A_1, A_2)$ theory.), and the later $ADE$ generalizations [99] (They are $(A_1, G)$ type theories) with $G = ADE$. This class of theories typically do not have any non-abelian flavor symmetries, although they could have abelian flavor symmetries.

2. Theories with label $D_p(G)$ [100], where $p$ is a positive integer and $G = ADE$. For $G = A_N$, they are called type IV theory in [58]. This class of theories has a flavor symmetry group $G$ and possibly some more abelian flavor symmetry depending on value of $p$.

3. Theories with label $(J^{(b)}[k], f)$ in [59], with $k > -b$. They were studied in [58,59] and are defined using 6d $(2, 0)$ SCFT with following data,
$$J = ADE, \;\; \Phi = \frac{T}{z^{2 + \frac{k}{b}}}, \;\; f. \tag{114}$$

   Here $f$ is a nilpotent orbit of $J = ADE$[10], and $T$ is a regular semi-simple matrix whose form depending on value $b$. $b$ takes a finite set of numbers as in table 1, and in particular $b$ can always take the value $h^\vee$ which is the dual Coxeter number. For $J = A_{N-1}, b = N$, it is called type I theory in [58], and for $b = N - 1$, it is called type II theory in [58].

We have the following mapping between the third class of theories and the first two class of theories:
$$(J^{h^\vee}[k], f_{reg}) = (J, A_{k-1}), \;\;\; (J^{h^\vee}[k], f_{trivial}) = D_{k+h^\vee}(J). \tag{115}$$
Here $h^\vee$ is the dual Coxeter number.

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
