# Peer review of "Schur sector of Argyres-Douglas theory and $W$-algebra"

_SciPost Physics, doi:SciPost Phys. 10, 080 (2021)_

## Round 1 · Referee Report · Anonymous (Referee 4) · 2021-2-13

Strengths
1-New results on the Schur index of 4d SCFTs
2-New observations on the structure of the corresponding quasi-lisse VOAs.
Weaknesses
-The general structure and grammar could be improved.
Report
This is an interesting paper. It explores the structure of the Schur operators of 4d N=2 SCFTs using the corresponding VOA, which is (conjecturally) quasi-lisse. They focus on SCFTs of class-S engineered with one irregular and one regular puncture, whose VOA are a some particular W-algebras (as shown by the authors in previous works).
The main new technical result of this paper is a explicit compact formula for the Schur index of these theories, computed as the vacuum character of the VOA, as a plethystic exponential. They then study Zhu's C_2 algebra for these VOAs, in a number of examples. In the case of a lisse VOAs (that is, for SCFTs without Higgs branch), the C_2 algebra is finitely generated and they observe that it can be written as the Jacobi algebra of an isolated singularity. (The significance of that interesting fact, if any, is not explained.) More generally, they conjecture that the algebra is always of the type C[X]/I, where the finite number of generators X can be identified from the Schur index, and I some ideal. They conclude with some interesting comments on the structure of the MacDonald index (a one-parameter refinement of the Schur index).
The paper appeared on the arXiv two years ago. Given that timeline, it is disappointing that the authors did not find it necessary to proofread their work and submit a more polished version to the arXiv before submitting to a journal. I point out some obvious typos in the requested changes below, but I would recommend a thorough proofreading, especially for improving the grammar.
The authors introduce some notation and facts without proper explanation nor referencing:
-on p16, the theories D_p(SU(N)) of Cecotti-del Zotto are introduced without explanation or reference. Similarly on p23.
-For (4.47), one should cite Shapere-Tachikawa.
Finally, it would have been useful to have discussed the relation to contemporaneous developments in the literature. My main question, as a non-expert, would be: How does the proposal for the "Schur ring" (identified as the C_2 algebra) as being given C[X]/I, for X a finite number of generators, relates to the free field description of Beem-Meneghelli-Rastelli (1903.07624)?
Requested changes
1-first line of section 2.3: "is always a lisse VOA": I believe you meant "is always a quasi-lisse VOA"
2-"principal" is wrongly spelled as "principle" 7 times.
3-Add references to Cecotti-Del Zotto and Shapere-Tachikawa.
4- Explain relation to 1903.07624.
Author: Wenbin Yan on 2021-02-14 [id 1236]
(in reply to Report 1 on 2021-02-13)Dear Sir/Madam,
Thank you very much for your comments and suggestions. We will improve our manuscript as you recommended.
For your question on the relation between Schur ring and the free field description, one gets a reduced ring from the Shur ring, which gives the coordinate ring of the Higgs branch. The coordinate ring of Higgs branch is related to the free field description. We will add more explanation in the manuscript.
Thanks again for your time and important opinions.
Best,
Wenbin

---

## Round 1 · Referee Report · Anonymous (Referee 3) · 2021-2-14

Weaknesses
1. The presentation is poor. It is hard to extract the results reading the paper.
Report
In my opinion the paper can be published in SciPost only after major revision.
This paper studies certain aspects of the Vertex Operators Algebras associated (in the sense of [5]) to four dimensional Argyres--Douglas superconformal field theories.
The main results are
A) An explicit form of the Schur index, see equation (1.1).
B) A proposal for the explicit description of the Zhu's $C_2$-algebra of some of the VOAs presented in the paper as the Jacobi algebra of
an auxiliary hypersurface singularity.
V) A proposal of identifying the Macdonald filtration (which comes from $4d$ data) with the Kazhdan filtration of the associated VOA.
Here is a list of point that should be addressed:
1. The presentation is often not very clear. This must be improved before publication.
For example is not clear which class of VOAs are treated in this paper.
In particular it should be stated in a more accessible way which triple $(\mathfrak{g},f,k')$ are considered.
2. Concerning equation (1.1), is it really new or it is a simple rewriting of the formulae presented in [62] which are already in a product form. The expression (1.1) makes it a little easier to check certain properties as done in section 4.3.1, but is it really crucial?
3. I think that section 2.1 would benefit from more references to the relevant math literature.
4. I believe that (4.12) and (5.1) contain a typo and that the numerator in the argument of $PE$ should be
\begin{equation*}
\sum_i q^{d_i} -q^{h^{\vee}+k+1}\sum _i q^{-d_i} \,.
\end{equation*}
This typo is unfortunate since this numerator contains important information about the generators and relations that describe the Zhu's $C_2$-algebra.
In Example 1, page 22, we have $\{d_1,d_2\}=\{2,3\}$, $\{h^{\vee}+k+1-d_1,h^{\vee}+k+1-d_2\}=\{6,5\}$. I also find that the terminology ``singular vector at level'' is confusing in this part and just referring to the conformal weight of the null or of the relations would be clearer.
5. Concerning the $C_2$-algebra, it would be useful to mention a few examples in which it has been computed explicitly and describe how it behaves under quantum Hamiltonian reduction.
Additionally it seems that the $C_2$-algebra does not have any sharp $4d$ meaning and I think this should be stated more clearly in point 2 in page 3.

---

## Round 2 · Referee Report · Anonymous (Referee 2) · 2021-3-10

Report

In my opinion the paper can be published as is.

---

## Round 2 · Referee Report · Anonymous (Referee 1) · 2021-3-18

Report

The authors addressed the points in the previous report. I recommend the paper for publication.

---

## Round 2 · List of Changes

Changes following suggestions of referee 1 1. corrected to quasi-lisse VOA 2. corrected all the typos including the ones pointed out by the referee 3. References are added accordingly. We also add an appendix explaining different names for generalized AD theories 4. Briefly explained the relation to 1903.07624. Actually more is discussed in our subsequent paper arXiv:1910.02281

Changes following suggestions of referee 2 1. corrections are made to make our results clear. For example, we stated clearly the triple (g,f,k') studied are for any simple Lie algebra g, any nilpotent orbit f, and boundary admissible level k'(which is also defined in the introduction) 2. We emphasis that although our PE formula is equivalent to the formula by mathematicians. Our form has the advantage that dualities are manifest in the PE form, and one can easily read off generators and singular vectors from the PE formula 3. added more references to math literature. 4. there are indeed typos and confusions in that part, and we corrected accordingly. 5. We made clear that right now the C2 algebra does not have a sharp 4d meaning. However, we still think it's worth study because not many 4d theories have no Higgs branch, but Schur sector is always there. Unfortunately we also don't have good description how C2 algebra behaves under quantum Hamiltonian reduction.

---

## Editorial Decision

published